# Data Scaling Laws in Imitation Learning for Robotic Manipulation

**Fanqi Lin**[1,2,3*]   **Yingdong Hu**[1,2,3*]   **Pingyue Sheng**[1]
**Chuan Wen**[1,2,3]   **Jiacheng You**[1]   **Yang Gao**[1,2,3]
[1]Tsinghua University, [2]Shanghai Qi Zhi Institute, [3]Shanghai Artificial Intelligence Laboratory
[*]Equal contribution    Project page: https://data-scaling-laws.github.io/

## Abstract

Data scaling has revolutionized fields like natural language processing and computer vision, providing models with remarkable generalization capabilities. In this paper, we investigate whether similar data scaling laws exist in robotics, particularly in robotic manipulation, and whether appropriate data scaling can yield single-task robot policies that can be deployed zero-shot for any object within the same category in any environment. To this end, we conduct a comprehensive empirical study on data scaling in imitation learning. By collecting data across numerous environments and objects, we study how a policy's generalization performance changes with the number of training environments, objects, and demonstrations. Throughout our research, we collect over 40,000 demonstrations and execute more than 15,000 real-world robot rollouts under a rigorous evaluation protocol. Our findings reveal several intriguing results: the generalization performance of the policy follows a roughly *power-law* relationship with the number of environments and objects. The *diversity* of environments and objects is far more important than the absolute number of demonstrations; once the number of demonstrations per environment or object reaches a certain threshold, additional demonstrations have minimal effect. Based on these insights, we propose an efficient data collection strategy. With four data collectors working for one afternoon, we collect sufficient data to enable the policies for two tasks to achieve approximately 90% success rates in novel environments with unseen objects.

## 1 Introduction

Scaling has been a key driver behind the rapid advancements in deep learning (Brown et al., 2020; Radford et al., 2021). In natural language processing (NLP) and computer vision (CV), numerous studies have identified scaling laws demonstrating that model performance improves with increases in *dataset size*, *model size*, and total *training compute* (Kaplan et al., 2020; Henighan et al., 2020). However, comprehensive scaling laws have not yet been established in robotics, preventing the field from following a similar trajectory. In this paper, we explore the first dimension of scaling—data—as scaling data is a prerequisite for scaling models and compute. We aim to investigate whether data scaling laws exist in robotics, specifically in the context of robotic manipulation, and if so, what insights and guidance they might offer for building large-scale robotic datasets.

While data scaling has endowed models in NLP and CV with exceptional generalization capabilities Achiam et al. (2023); Kirillov et al. (2023), most of today's robotic policies still lack comparable zero-shot generalization (Xie et al., 2024). From the outset, we treat generalizable manipulation skills as first-class citizens, emphasizing real-world generalization over evaluations in controlled lab settings. In this context, we aim to investigate the following fundamental question: *Can appropriate data scaling produce robot policies capable of operating on nearly **any** object within the same category, in **any** environment?*

To answer this, we present a comprehensive empirical study on data scaling in imitation learning, which is a predominant method for learning real-world manipulation skills (Shafiullah et al., 2024). We categorize generalization into two dimensions: *environment generalization* and *object generalization*, which essentially encompass all factors a policy may encounter during real-world

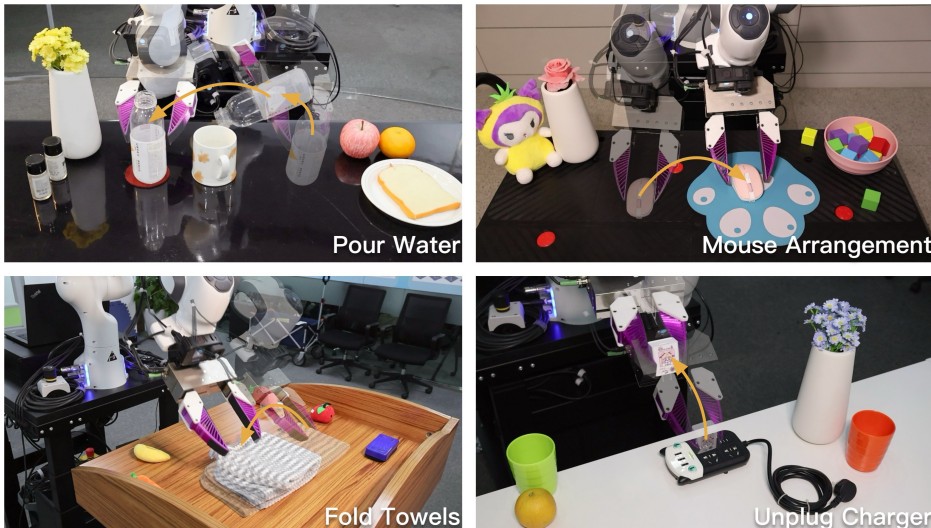

Figure 1: **Illustrations of all tasks.** We derive the data scaling laws through extensive experiments on `Pour Water` and `Mouse Arrangement`, and further validate these findings on additional tasks, including `Fold Towels` and `Unplug Charger`.

deployment. We do not consider task-level generalization at this stage, as we believe it would require collecting vast amounts of data from thousands of tasks (Padalkar et al., 2023; Khazatsky et al., 2024), which is beyond the scope of our work. Instead, we systematically explore how a single-task policy's performance changes in new environments or with new objects as the number of training environments or objects increases. Additionally, we examine how the number of demonstrations impacts policy generalization when the number of environments and objects is fixed.

We use hand-held grippers (i.e., UMI (Chi et al., 2024)) to collect human demonstrations in various environments and with different objects, modeling this data using a Diffusion Policy (Chi et al., 2023) (Sec. 3). We begin by focusing on two tasks as case studies—`Pour Water` and `Mouse Arrangement`—to thoroughly analyze how policy generalization changes with the number of environments, objects, and demonstrations (Sec. 4.1), summarizing data scaling laws (Sec. 4.2). Then, based on these data scaling laws, we propose an efficient data collection strategy to achieve the desired level of generalization (Sec. 4.3). We apply this strategy to two new tasks (`Fold Towels` and `Unplug Charger`), and within a single afternoon using four data collectors, we collect sufficient data to train policies that achieve around 90% success rates across 8 new environments and objects for each task (Sec. 5). Lastly, we go beyond data scaling by conducting preliminary explorations of model size scaling (Sec. 6). Throughout our research, we collect over 40,000 demonstrations and conduct all experiments under a rigorous evaluation protocol that included more than 15,000 real-world robot rollouts. Our extensive investigation reveals surprising results and contributions:

- **Simple power laws.** The policy's generalization ability to new objects, new environments, or both scales approximately as a power law with the number of training objects, training environments, or training environment-object pairs, respectively.

- **Diversity is all you need.** Increasing the diversity of environments and objects is far more effective than increasing the absolute number of demonstrations per environment or object.

- **Generalization is easier than expected.** Collecting data in as many environments as possible (e.g., 32 environments), each with one unique manipulation object and 50 demonstrations, allows training a policy that generalizes well (90% success rate) to any new environment and new object.

## 2    RELATED WORK

**Scaling laws.** Scaling laws are first discovered in neural language models (Kaplan et al., 2020), revealing a power-law relationship between dataset size (or model size, computation) and cross-entropy loss. Subsequently, scaling laws have been observed in discriminative image modeling (Zhai

et al., 2022), generative image modeling (Peebles & Xie, 2023), video modeling (Henighan et al., 2020), and other domains (Hilton et al., 2023; Liu et al., 2024). These laws not only validate the scalability of neural networks—a key factor in the success of recent foundation models (Bommasani et al., 2021; Brown et al., 2020; Touvron et al., 2023)—but also enable performance prediction for larger models based on their smaller counterparts, thereby guiding more effective resource allocation (Achiam et al., 2023). In this paper, we examine data scaling laws to explore the relationship between the generalization of robot policies and the number of environments, objects, and demonstrations, and to develop efficient data collection strategies based on these insights.

**Data scaling in robotic manipulation.** Similar to the fields of NLP and CV, robotic manipulation is also experiencing a trend toward scaling up data (Sharma et al., 2018; Kalashnikov et al., 2018; Mandlekar et al., 2018; Dasari et al., 2019; Ebert et al., 2021; Jang et al., 2022; Brohan et al., 2022; Walke et al., 2023; Bharadhwaj et al., 2023; Fang et al., 2023a; Shafiullah et al., 2023; Padalkar et al., 2023; Khazatsky et al., 2024; Zhao et al., 2024). The largest existing dataset, Open X-Embodiment (OXE) (Padalkar et al., 2023), comprises over 1 million robot trajectories from 22 robot embodiments. The primary objective of scaling OXE is to develop a foundational robot model that facilitates positive transfer learning across different robots. However, deploying such models in new environments still requires data collection for fine-tuning. In contrast, our scaling objective focuses on training a policy that can be directly deployed in novel environments and with unseen objects, eliminating the need for fine-tuning. Additionally, we observe that Gao et al. (2024) also explore strategies for efficient data scaling to enhance generalization. However, their work is limited to *in-domain* compositional generalization, whereas our focus is on *out-of-domain* generalization.

**Generalization in robotic manipulation.** Creating a generalizable robot has been a longstanding aspiration within the robotics community. Some research aims to improve generalization to new object instances (Mahler et al., 2017; Mu et al., 2021; Fang et al., 2023b; Zhu et al., 2023a), while other efforts focus on enabling robots to adapt to unseen environments (Hansen et al., 2020; Xing et al., 2021; Teoh et al., 2024; Xie et al., 2024). Recently, significant attention has been paid to developing policies that can generalize to new task instructions (Jang et al., 2022; Bharadhwaj et al., 2023; Brohan et al., 2023; Team et al., 2024). In this paper, we concentrate on the first two dimensions of generalization: creating a single-task policy capable of operating on nearly *any object* within the same category, in *any environment*. This kind of single-task policy can serve as a primitive skill for planning algorithms (Ahn et al., 2022; Hu et al., 2023a) and is also the foundation for further research into multi-task generalist policies (Kim et al., 2024). UMI (Chi et al., 2024) demonstrates that training on diverse demonstrations significantly enhances the generalization performance of policies in novel environments and with novel objects. Concurrently with our work, RUMs (Etukuru et al., 2024) develop policies capable of zero-shot deployment in novel environments. However, neither UMI nor RUMs delves into a comprehensive analysis of the relationship between generalization and different data dimensions—a gap our work aims to address.

## 3 APPROACH

In this section, we first outline the generalization dimensions we consider and the formal formulation of the data scaling laws. Then, we demonstrate our data source and design choices for policy learning methods. Finally, we introduce our rigorous evaluation protocol.

**Generalization dimensions.** We use behavior cloning (BC) to train single-task policies, a dominant approach for learning real-world manipulation skills. However, many BC-trained policies exhibit poor generalization performance. This generalization issue manifests across two dimensions: (1) *Environment*—generalization to previously unseen environments, which may involve variations in lighting conditions, distractor objects, background changes, and more; (2) *Object*—generalization to new objects within the same category as those in human demonstrations, differing in attributes such as color, size, geometry, and so on.

Prior research in this area has attempted to isolate the variations within each dimension by controlling specific factors independently (Xie et al., 2024; Pumacay et al., 2024). For instance, special lighting setups might be used to change only the color of illumination, or 3D-printed objects might be designed to vary only in size without altering their shape or geometry. While this approach allows precise control over individual factors, it cannot account for all possible variation factors. More importantly, real-world performance depends not on generalizing to individual factors but on handling

the complex interplay of multiple factors that vary simultaneously. To address this, we focus on generalization across two dimensions—*environment* and *object*—which collectively encompass all factors a policy may encounter in natural, real-world scenarios. For environment variations, we scale the number of real scenes by collecting human demonstrations across diverse in-the-wild environments. For object variations, we scale the number of accessible objects by acquiring a large variety of everyday items within the same category. See Appendix A for visualizations of the environments and objects used in our study. We believe that this emphasis on real-world diversity enhances the applicability of our findings to more varied and practical contexts.

**Data scaling laws formulation.** For simplicity, we consider a scenario where a demonstration dataset for a manipulation task is collected across $M$ environments $(E_1, E_2, \ldots, E_M)$ and $N$ manipulation objects of the same category $(O_1, O_2, \ldots, O_N)$. Each environment may contain any number of distractor objects, provided they are not in the same category as the manipulation objects. For each object $O_i$ in an environment $E_j$, $K$ demonstrations $(D_{ij1}, D_{ij2}, \ldots, D_{ijK})$ are collected. We evaluate the policy's performance using test scores $S$ (described in detail later) on environments and objects not seen during training. The data scaling laws in this paper aim to: (1) characterize the relationship between $S$ and the variables $M$, $N$, and $K$, specifically, how the generalization ability depends on the number of environments, objects, and demonstrations; and (2) determine efficient data collection strategies to achieve the desired level of generalization based on this relationship.

**Data source.** Existing robotic manipulation datasets do not provide enough environments and objects for a single task to meet our requirements. Therefore, we opt to use the Universal Manipulation Interface (UMI) (Chi et al., 2024), a hand-held gripper, to independently collect a substantial number of demonstrations. UMI's portability, intuitive design, and low cost make it an ideal tool for our data collection needs. It enables highly efficient data collection and allows for seamless switching between different in-the-wild environments with minimal setup time. However, as UMI relies on SLAM for capturing end-effector actions, it may encounter challenges in texture-deficient environments. We observe that approximately 90% of our collected demonstrations are valid. For more details on our data collection and experience with UMI, see Appendix B.

**Policy learning.** We employ Diffusion Policy to model the extensive data we collect, due to its demonstrated excellence in real-world manipulation tasks and its recent widespread application (Shafiullah et al., 2024; Ze et al., 2024). Following Chi et al. (2023), we utilize a CNN-based U-Net (Ronneberger et al., 2015) as the noise prediction network and employ DDIM (Song et al., 2020a) to reduce inference latency, achieving real-time control. See Appendix C for more training details. To further enhance performance, we make two improvements:

(1) DINOv2 visual encoder: In our experiments, fine-tuning the DINOv2 ViT (Oquab et al., 2023) outperforms both ImageNet pre-trained ResNet (He et al., 2016; Deng et al., 2009) and CLIP ViT (Radford et al., 2021). We attribute this improvement to DINOv2 features' ability to explicitly capture scene layout and object boundaries within an image (Caron et al., 2021). This information is crucial for enhanced spatial reasoning, which is particularly beneficial for robot control (Hu et al., 2023b; Yang et al., 2023; Kim et al., 2024). To ensure model capacity does not become a bottleneck when scaling data, we utilize a sufficiently large model, ViT-Large/14 (Dosovitskiy et al., 2020).

(2) Temporal ensemble: Diffusion Policy predicts a sequence of actions every $T_1$ steps, with each sequence having a length of $T_2$ $(T_2 > T_1)$, and only the first $T_1$ steps are executed. We observe that discontinuities between executed action sequences can cause jerky motions during switching. To address this, we implement the temporal ensemble strategy proposed in ACT (Zhao et al., 2023). Specifically, the policy predicts at each timestep, resulting in overlapping action sequences. At any given timestep, multiple predicted actions are averaged using an exponential weighting scheme, smoothing transitions and reducing motion discontinuity.

**Evaluation.** We conduct rigorous evaluations to ensure the reliability of our results. First, to evaluate the generalization performance of the policy, we exclusively test it in *unseen* environments or with *unseen* objects. Second, we use tester-assigned scores as the primary evaluation metric. Each manipulation task is divided into several stages or steps (typically 2–3), each with well-defined scoring criteria (see Appendix D). Each step can receive a maximum of 3 points, and we report a normalized score, defined as Normalized score $= \frac{\text{Total test score}}{3 \times \text{Number of steps}}$, with a maximum value of 1. Unlike the commonly used success rate—which is an overly sparse signal lacking the granularity to distinguish between policies—our scoring mechanism captures more nuanced behaviors. While action

mean squared error (MSE) on the validation set is another potential metric, we find it often does not correlate with real-world performance (see Appendix E.1 for more details). Finally, to minimize the tester's subjective bias, we simultaneously evaluate multiple policies trained on datasets of different sizes; each rollout is randomly selected from these multiple policies, while ensuring identical initial conditions for both the objects and the robot arm, enabling a fair comparison across policies. See Appendix E.2 for an example of the evaluation workflow and Appendix F for the hardware setup.

## 4 UNVEILING OF DATA SCALING LAWS

In this section, we first explore how increasing the number of training objects affects object generalization. Next, we analyze how the number of training environments impacts environment generalization. Finally, we study generalization across both dimensions simultaneously. Throughout all experiments, we also analyze the effect of demonstration quantity (Sec. 4.1). From these results, we derive the power-law data scaling laws (Sec. 4.2). Based on these laws, we further demonstrate an efficient data collection strategy to achieve a generalizable policy (Sec. 4.3).

### 4.1 RESULTS AND QUALITATIVE ANALYSIS

**Tasks.** We first focus on two manipulation tasks: `Pour Water` and `Mouse Arrangement`. In `Pour Water`, the robot performs three steps: first, it grabs a drinking bottle placed randomly on the table; second, it pours water into a mug; and finally, it places the bottle on a red coaster. This task demands precision, especially in aligning the bottle's mouth with the mug. In `Mouse Arrangement`, the robot completes two steps: it picks up a mouse and positions it on a mouse pad with its front facing forward. The mouse may be tilted, requiring the robot to employ non-prehensile actions (i.e., pushing) to first align it. Illustrations of all tasks are shown in Fig. 1, with further task details available in Appendix D. Robot rollout videos can be found on our website.

**Object generalization.** We use 32 distinct objects within the same environment to collect 120 demonstrations per object, yielding a total of 3,840 demonstrations for each task. After SLAM filtering, the number of valid demonstrations for `Pour Water` and `Mouse Arrangement` is reduced to 3,765 and 3,820, respectively. To investigate how the number of training objects influences the policy's ability to generalize to unseen objects, we randomly select $2^m$ objects ($m = 0, 1, 2, 3, 4, 5$) from the pool of 32 for training. Furthermore, to examine how policy performance varies with the number of demonstrations, we randomly sample $2^n$ fractions of valid demonstrations ($n = 0, -1, -2, -3, -4, -5$) for each selected object. For each combination of $(m, n)$, we train a policy if the total number of demonstrations exceeds 100. In total, 21 policies are trained, and each is evaluated using 8 *unseen objects* in the same environment as the training data, with 5 trials per object. The average normalized score across 40 trials is reported for each policy.

Fig 2 presents the results, with shaded regions representing 95% confidence intervals. There are several key observations: (1) As the number of training objects increases, the policy's performance on unseen objects consistently improves across all fractions of demonstrations. (2) With more training objects, fewer demonstrations are required per object. For example, in `Pour Water`, when training with 8 objects, the performance using 12.5% of the demonstrations significantly lags behind that using 100% of the demonstrations; however, this gap nearly disappears when training with 32 objects. (3) Object generalization is relatively easy to achieve. The initial slope of the performance curve is very steep: with only 8 training objects, the normalized score for both tasks exceeds 0.8. When the number of training objects reaches 32, the score surpasses 0.9. These scores correspond to policies that have already generalized well to any new objects within the same category.

**Environment generalization.** To explore the effect of the number of training environments on generalization, we use the same manipulation object across 32 distinct environments, collecting 120 demonstrations per environment. For `Pour Water` and `Mouse Arrangement`, this result in 3,424 and 3,351 valid demonstrations, respectively. We randomly select $2^m$ environments ($m = 0, 1, 2, 3, 4, 5$) from the 32 available for training, and for each selected environment, we randomly select $2^n$ fractions of vaild demonstrations ($n = 0, -1, -2, -3, -4, -5$). Each policy is evaluated in 8 *unseen environments* using the same object as in training, with 5 trials per environment.

Fig. 3 presents the results, revealing several notable patterns: (1) Increasing the number of training environments enhances the policy's generalization performance on unseen environments. This trend

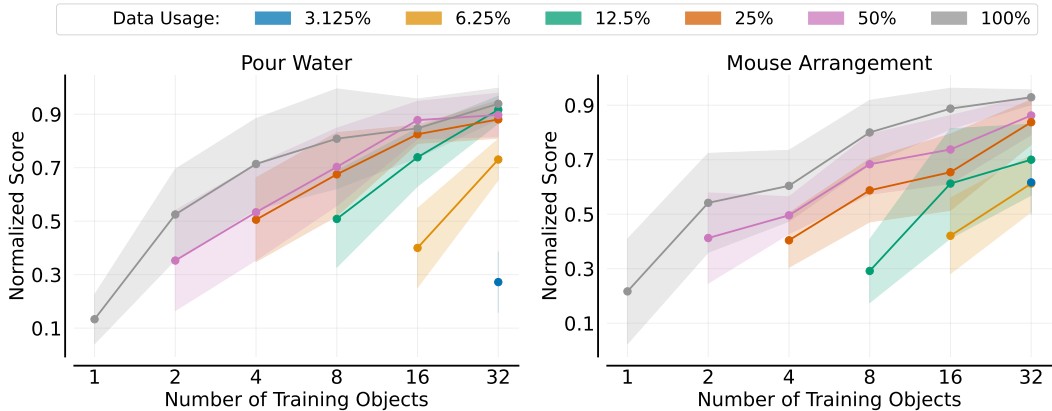

Figure 2: **Object generalization.** Each curve corresponds to a different fraction of demonstrations used, with normalized scores shown as a function of the number of training objects.

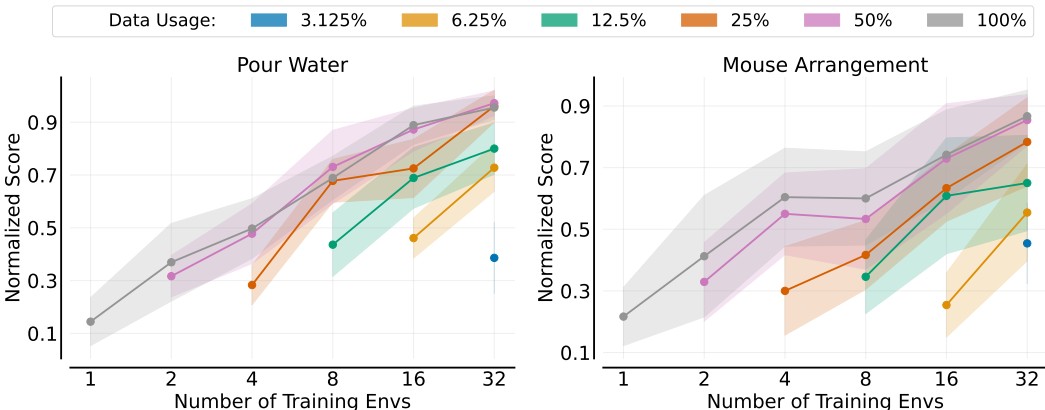

Figure 3: **Environment generalization.** Each curve corresponds to a different fraction of demonstrations used, with normalized scores shown as a function of the number of training environments.

persists even when the total number of demonstrations is kept constant (see Appendix G.2, Fig. 22). However, while increasing the fraction of demonstrations in each environment initially boosts performance, this improvement quickly diminishes, as indicated by the overlap of the lines representing 50% and 100% demonstration usage. (2) Environment generalization appears to be more challenging than object generalization for these two tasks. Comparing Fig. 2 and Fig. 3, we observe that when the number of environments or objects is small, increasing the number of environments results in smaller performance gains compared to increasing the number of objects. This is reflected in the lower slope of the performance curve for environment generalization.

**Generalization across both environments and objects.** Next, we explore a setting where both the training environments and objects vary simultaneously. Data is collected from 32 environments, each paired with a unique object. For `Pour Water` and `Mouse Arrangement`, the number of valid demonstrations is 3,648 and 3,564, respectively. We randomly select $2^m$ environment-object pairs ($m = 0, 1, 2, 3, 4, 5$) from the pool of 32 for training and, for each selected pair, we randomly sample $2^n$ fractions of valid demonstrations ($n = 0, -1, -2, -3, -4, -5$). Each policy is evaluated in 8 *unseen environments*, using two *unseen objects* per environment, with 5 trials per environment.

Fig. 4 illustrates that (1) increasing the number of training environment-object pairs substantially enhances the policy's generalization performance, consistent with previous observations. (2) Interestingly, although generalizing across both novel environments and objects is more challenging, the benefit of additional demonstrations saturates faster in such cases (as evidenced by the overlapping lines for 25% and 100% demonstration usage). This indicates that, compared to changing either the environment or the object alone, simultaneously changing both increases data *diversity*, leading to more efficient policy learning and reducing dependence on the number of demonstrations. This find-

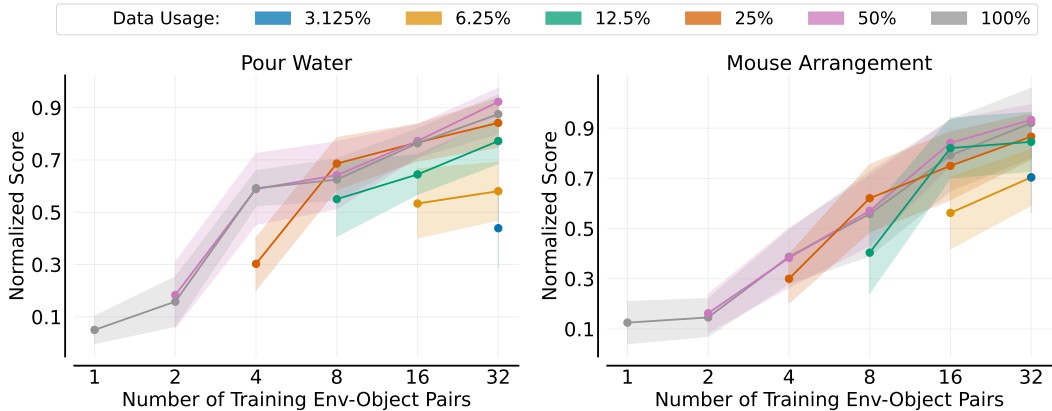

Figure 4: **Generlization across environments and objects.** Each curve corresponds to a different fraction of demonstrations used, with normalized scores shown as a function of the number of training environment-object pairs.

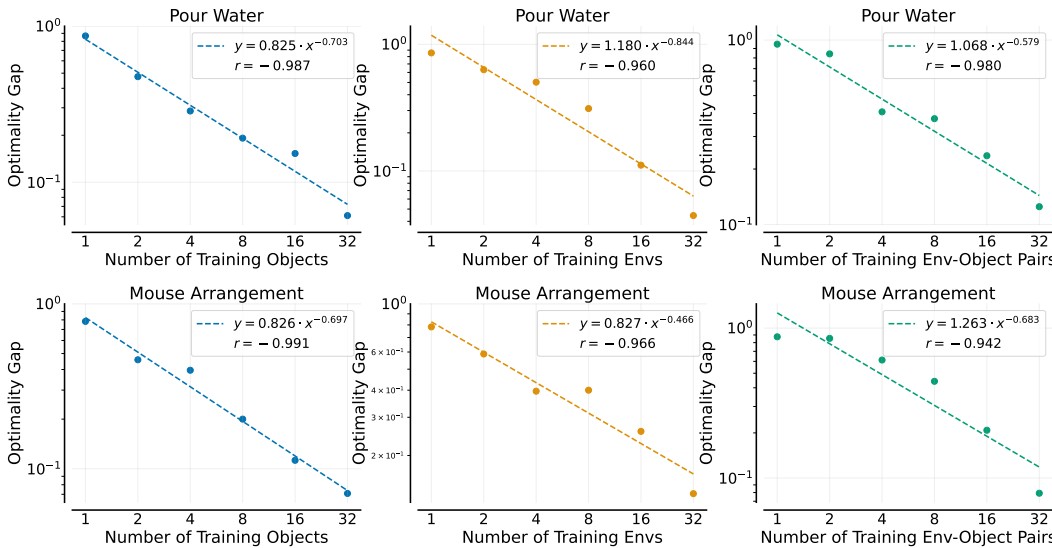

Figure 5: **Power-law relationship.** Dashed lines represent power-law fits, with the equations provided in the legend. All axes are shown on a logarithmic scale. The correlation coefficient (Pearson's $r$) indicates a power-law relationship between the generalization ability and the number of objects, environments, and environment-object pairs. See Appendix G.1 for data scaling laws on MSE.

ing further emphasizes that expanding the diversity of environments and objects is more effective than merely increasing the number of demonstrations for each individual environment or object.

## 4.2 POWER-LAW FITTING AND QUANTITATIVE ANALYSIS

We next explore whether our experimental results follow power-law scaling laws, as seen in other domains. Specifically, if two variables $Y$ and $X$ satisfy the relation $Y = \beta \cdot X^{\alpha}$[1], they exhibit a power-law relationship. Applying a logarithmic transformation to both $Y$ and $X$ reveals a linear relationship: $\log(Y) = \alpha \log(X) + \log(\beta)$. In our context, $Y$ represents the optimality gap, defined as the deviation from the maximum score (i.e., $1 -$ Normalized Score), while $X$ can denote the number of environments, objects, or demonstrations. Using data from our previous experiments with a 100% fraction of demonstrations, we fit a linear model to the log-transformed data, as shown in Fig. 5. Based on all the results, we summarize the following data scaling laws:

---

[1]Although we recognize the irreducible errors $Y_{\infty}$ associated with scaling the data alone, fitting a three-parameter model $Y = \beta X^{\alpha} + Y_{\infty}$ does not seem statistically justified given that we have only 6 data points.

- The policy's generalization ability to new objects, new environments, or both scales approximately as a *power law* with the number of training objects, training environments, or training environment-object pairs, respectively. This is evidenced by the correlation coefficient $r$ in Fig. 5.

- When the number of environments and objects is fixed, there is no clear power-law relationship between the number of demonstrations and the policy's generalization performance. While performance initially increases rapidly with more demonstrations, it eventually plateaus, as most clearly shown in the leftmost plot of Fig. 7 (see caption for details).

These power laws regarding environments and objects can serve as predictive tools for larger-scale data. For example, according to the equation in Fig. 5, we predict that for `Mouse Arrangement`, achieving a normalized score of 0.99 on novel environments and objects would require 1,191 training environment-object pairs. We leave the verification of this prediction for future work.

## 4.3 Efficient Data Collection Strategy

In this section, we present an efficient data collection strategy guided by the data scaling law. Recall that our data is collected across $M$ environments and $N$ manipulation objects, with $K$ demonstrations for each object in every environment. The main question we seek to answer is: for a given manipulation task, how can we optimally select $M$, $N$, and $K$ to ensure strong generalization of the policy without incurring an excessively laborious data collection process? To explore this, we continue to use the tasks `Pour Water` and `Mouse Arrangement` as examples.

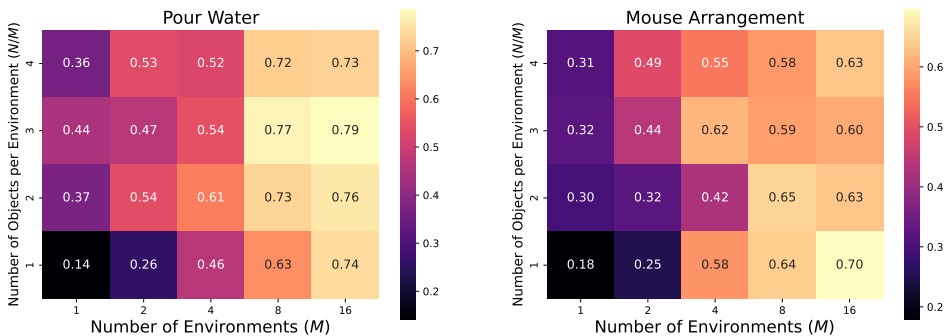

Figure 6: **Multiple objects per environment.** Brighter colors indicate higher normalized scores.

**How to select the number of environments and objects?** Previously, we consider only the setting where each environment contains a single unique manipulation object. In practical data collection, however, collecting multiple objects per environment might improve performance and thus be a more efficient method. To explore this possibility, we assume that $N$ is a multiple of $M$, with each environment containing $N/M$ unique objects. Specifically, we use 16 environments, each containing 4 unique objects, and collect 120 demonstrations for each object ($M = 16, N = 64, N/M = 4, K = 120$). For `Pour Water` and `Mouse Arrangement`, this results in 6,896 and 6,505 valid demonstrations, respectively. We then randomly select $2^m$ environments ($m = 0, 1, 2, 3, 4$) from the 16 available environments. For each selected environment, we use all demonstrations of $n$ objects ($n = 1, 2, 3, 4$) as the training data. In total, we train 20 policies, each evaluated in 8 unseen environments using two novel objects per environment, with 5 trials for each environment.

The heatmap in Fig. 6 shows that when the number of environments is small, collecting multiple objects in each environment boosts performance. However, as the number of environments increases (e.g., to 16), the performance gap between collecting multiple objects per environment and just a single object becomes negligible. For large-scale data collection, where the number of environments typically exceeds 16, adding multiple objects within the same environment does not further enhance policy performance, suggesting that this approach may be unnecessary. Based on our experimental results, we recommend the following: *collect data in as many diverse environments as possible, with only one unique object in each environment.* When the total number of environment-object pairs reaches 32, it is generally sufficient to train a policy capable of operating in novel environments and interacting with previously unseen objects.

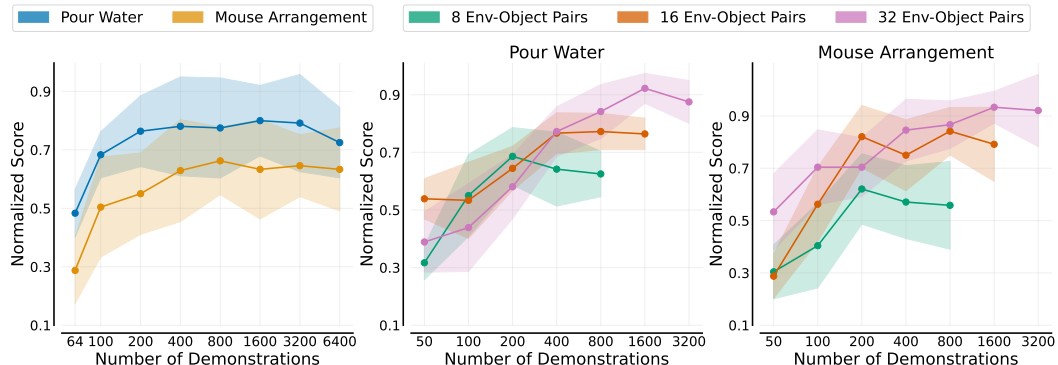

Figure 7: **Number of demonstrations. Left:** In the setting where we collect the maximum number of demonstrations, we examine whether the policy's performance follows a power-law relationship with the total number of demonstrations. The correlation coefficients for `Pour Water` and `Mouse Arrangement` are $-0.62$ and $-0.79$, respectively, suggesting only a weak power-law relationship. **Right:** For varying environment-object pairs, the policy performance increases with the total number of demonstrations at first, and then reaches saturation.

**How to select the number of demonstrations?** The experimental results in Sec. 4.1 indicate that increasing the number of demonstrations beyond a certain point yields minimal benefits. This section aims to identify that threshold. We first examine the setting where $M = 16$ and $N = 64$ (as in the previous experiment), representing the scenario with the maximum number of collected demonstrations—over 6400 in total. We vary the total number of demonstrations used for training, ranging from 64 to 6400, and train 8 policies. The results, presented in the leftmost plot of Fig. 7, show that performance for both tasks plateaus when the number of demonstrations reaches 800. Next, we consider our recommended setting of collecting environment-object pairs (i.e., $M = N$). The results, depicted in the two rightmost plots of Fig. 7, indicate that when the number of environment-object pairs is smaller, fewer total demonstrations are needed to reach saturation. Specifically, for 8, 16, and 32 pairs, performance plateaus at 400, 800, and 1600 demonstrations, respectively. Based on these findings, we recommend *collecting 50 demonstrations per environment-object pair (i.e., $K = 50$) for tasks of similar difficulty to ours*. More challenging dexterous manipulation tasks may require more demonstrations; we leave the exploration of this aspect to future work.

## 5 VERIFICATION OF DATA COLLECTION STRATEGY

|  | Pour Water | Mouse Arrangement | Fold Towels | Unplug Charger |
|---|---|---|---|---|
| Score | $0.922 \pm 0.075$ | $0.933 \pm 0.088$ | $0.95 \pm 0.062$ | $0.887 \pm 0.14$ |
| Success Rate | $85.0 \pm 19.4\%$ | $92.5 \pm 9.7\%$ | $87.5 \pm 17.1\%$ | $90.0 \pm 14.1\%$ |

Table 1: **Success rate across all tasks.** We report the average success rate and standard deviation across 8 unseen environments. The performance in each environment is detailed in Table 12.

To verify the general applicability of our data collection strategy, we apply it to new tasks and assess whether a sufficiently generalizable policy can be trained. We experiment with two new tasks: `Fold Towels` and `Unplug Charger`. In `Fold Towels`, the robot first grasps the left edge of the towel and folds it to the right. In `Unplug Charger`, the robot grabs the charger plugged into a power strip and swiftly pulls it out. For each task, we collect data from 32 environment–object pairs, with 50 demonstrations per environment. Consistent with previous experiments, we evaluate the policy in 8 unseen environments, each with 2 unseen objects, and perform 5 trials per environment. The results, shown in Table 1, report both the policy's normalized score and the corresponding success rate (for the definition of success criteria, see Appendix D). As the table indicates, our policies achieve around 90% success rates across all four tasks—the two from previous experiments and the two new ones. Notably, achieving this strong generalization performance on the two new tasks requires only one afternoon of data collection by four data collectors. This highlights the high efficiency of our data collection strategy and suggests that the time and cost required to train a single-task policy capable of zero-shot deployment to new environments and objects are moderate.

| Case | Score |
|------|-------|
| DINOv2 ViT-L/14 | **0.90** |
| LfS ViT-L/14 | 0.03 |
| frozen DINOv2 | 0.00 |
| LoRA DINOv2 | 0.72 |

(a) **Training strategy**. Both pre-training and full fine-tuning are indispensable.

| Case | Score |
|------|-------|
| DINOv2 ViT-S/14 | 0.66 |
| DINOv2 ViT-B/14 | 0.81 |
| DINOv2 ViT-L/14 | **0.90** |

(b) **Visual encoder scaling**. Scaling visual encoder yields a consistent performance boost.

| Case | Score |
|------|-------|
| small U-Net | 0.88 |
| base U-Net | **0.90** |
| large U-Net | 0.83 |

(c) **Diffusion model scaling**. Scaling action diffusion model does not bring a performance boost.

Table 2: **Model related experiments** on `Pour Water`. The entries marked in `gray` are the same, which specify the default settings: the visual encoder is a fully fine-tuned ViT-L/14 model pre-trained with DINOv2, while the action diffusion model employs a base-size 1D CNN U-Net.

## 6 MODEL SIZE AND TRAINING STRATEGY: BEYOND DATA SCALING

Finally, we extend our exploration beyond data scaling to investigate the model side. The Diffusion Policy consists of two components: a visual encoder and an action diffusion model. Our investigation focuses on the importance of the training strategy for the visual encoder and the effects of scaling the parameters of both the visual encoder and the action diffusion model. We conduct experiments on `Pour Water`, using data collected from 32 environment-object pairs and selecting 50% of all valid demonstrations as the training set. The results, shown in Table 2, lead to several key observations: (1) Both pre-training and full fine-tuning are essential for the visual encoder. As shown in Table 2a, a Learning-from-Scratch (LfS) ViT-L/14 and the use of frozen DINOv2 pre-trained features achieve scores close to *zero*. Additionally, parameter-efficient fine-tuning methods like LoRA (rank=8) (Hu et al., 2021) do not match the performance of full fine-tuning. (2) Increasing the size of the visual encoder significantly enhances performance. Table 2b demonstrates that scaling the visual encoder from ViT-Small to ViT-Large leads to a steady improvement in the policy's generalization performance. (3) Contrary to expectations, scaling the action diffusion U-Net does not yield performance improvements. As shown in Table 2c, despite the increase in maximum feature dimensions—from 512 to 2048—as the network scales from small to large, there is no corresponding improvement in score. In fact, performance slightly declines with the largest U-Net. We hypothesize that the small U-Net's capacity may already be sufficient for modeling the current action distribution, or that we have yet to identify a scalable architecture or algorithm for action diffusion.

## 7 DISCUSSION, LIMITATIONS, & FUTURE WORKS

Data scaling is an exciting and ongoing event in robotics. Rather than blindly increasing data quantity, emphasis should be placed on data quality. What types of data should be scaled? How can this data be efficiently obtained? These are the fundamental questions we aim to answer. Specifically, in the context of imitation learning, we uncover the significant value of diversity in environments and objects within human demonstrations, identifying a power-law relationship in the process. Furthermore, we believe that in-the-wild generalization is the ultimate goal of data scaling, and our study aims to demonstrate that this goal is closer than it may appear. We show that, with a relatively modest investment of time and resources, it is possible to learn a single-task policy that can be deployed zero-shot to any environment and object. To further support researchers in this endeavor, we release our code, data, and models, with the hope of inspiring further efforts in this direction and ultimately leading to general-purpose robots capable of solving complex, open-world problems.

Our work has several limitations that future research can address. First, we focus on data scaling for single-task policies and do not explore task-level generalization, as this would require collecting data from thousands of tasks. Future studies could incorporate language-conditioned policies to explore how to scale data to obtain a policy that can follow any new task instructions (Kim et al., 2024). Second, we study data scaling only in imitation learning, while reinforcement learning (RL) likely enhances policy capabilities further; future research can investigate the data scaling laws for RL. Third, our use of UMI for data collection introduces inherent small errors in the demonstrations, and we model the data using only Diffusion Policy algorithm. Future research can investigate how data quality and policy learning algorithms affect data scaling laws. Lastly, due to resource constraints, we explore and validate data scaling laws on only four tasks; we hope that future work will verify our conclusions on a larger and more complex set of tasks.

ACKNOWLEDGMENTS

This work is supported by the National Key R&D Program of China (2022ZD0161700), National Natural Science Foundation of China (62176135), Shanghai Qi Zhi Institute Innovation Program SQZ202306 and the Tsinghua University Dushi Program.

The robot hardware is partially supported by Tsinghua ISR Lab. We would like to express our gratitude to Cheng Chi and Chuer Pan for their invaluable advice on UMI. We are also thankful to Linkai Wang for his assistance in setting up the movable platform. Additionally, we appreciate the thoughtful discussions and feedback provided by Tong Zhang, Ruiqian Nai, Geng Chen, Weijun Dong, Shengjie Wang, and Renhao Wang.

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

APPENDICES

# A  ENVIRONMENT AND OBJECT VISUALIZATIONS

## A.1  ENVIRONMENT VISUALIZATIONS

Figures 8,9,10, and 11 present the sampled training environments for each of the four tasks.

Figure 12 presents the 8 unseen testing environments.

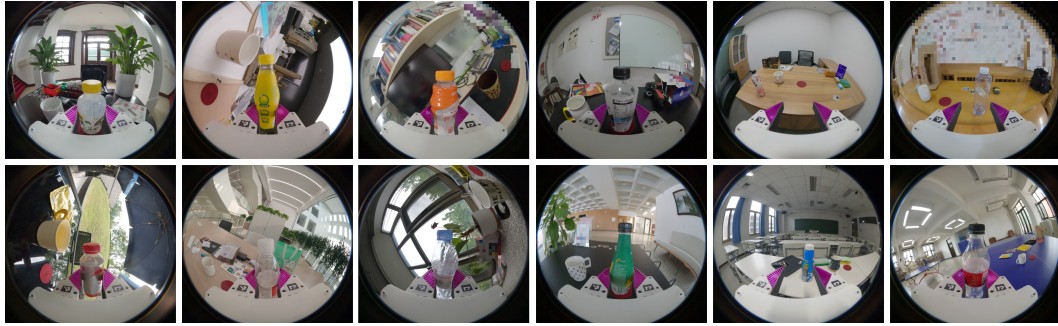

Figure 8: **Training environments for `Pour Water`.** We sample 12 environments from our collected training data. See Appendix D.1 for task details.

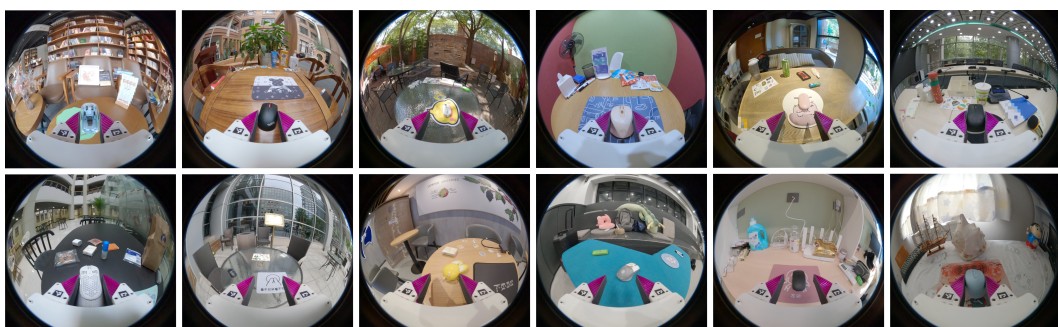

Figure 9: **Training environments for `Mouse Arrangement`.** We sample 12 environments from our collected training data. See Appendix D.2 for task details.

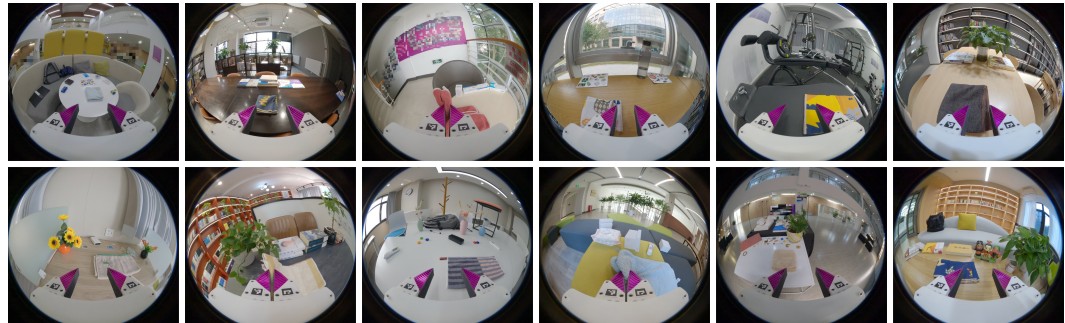

Figure 10: **Training environments for `Fold Towels`.** We sample 12 environments from our collected training data. See Appendix D.3 for task details.

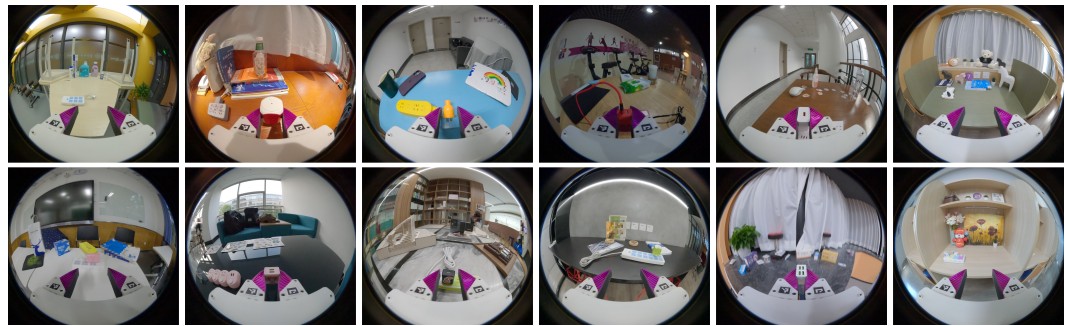

Figure 11: **Training environments for `Unplug Charger`.** We sample 12 environments from our collected training data. See Appendix D.4 for task details.

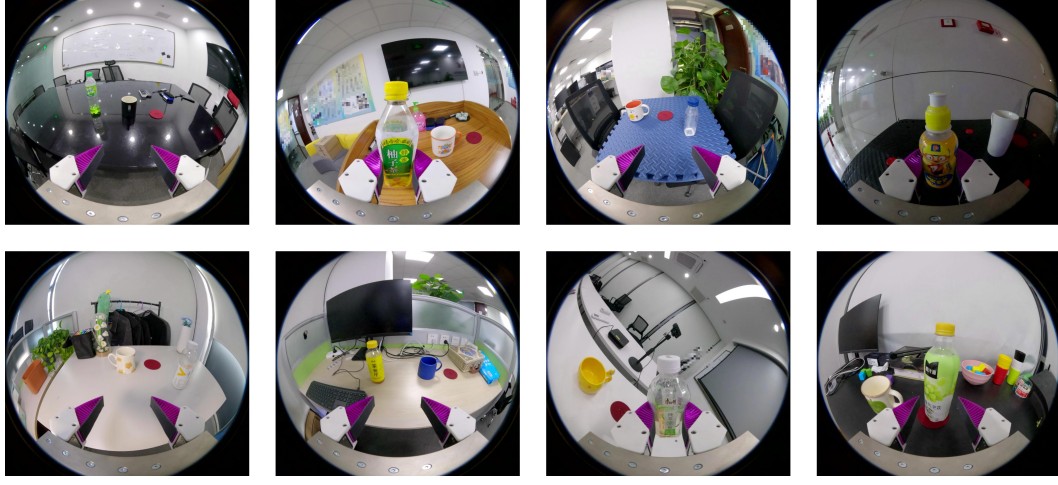

Figure 12: **Testing environments.** These 8 environments are not included in the training data and are used across all tasks.

A.2 OBJECT VISUALIZATIONS

In Figures 13, 14, 15, and 16, we present the training and testing objects for the tasks Pour Water, Mouse Arrangement, Fold Towels, and Unplug Charger, respectively.

Note that when we refer to "one manipulation object" in a task, we are actually referring to all the objects involved in completing that task. For instance, in Pour Water, this includes both the drink bottle and the mug. In Mouse Arrangement, it refers to the mouse and the mouse pad. In Fold Towels, it applies solely to the towel. In Unplug Charger, it encompasses both the charger and the power strip.

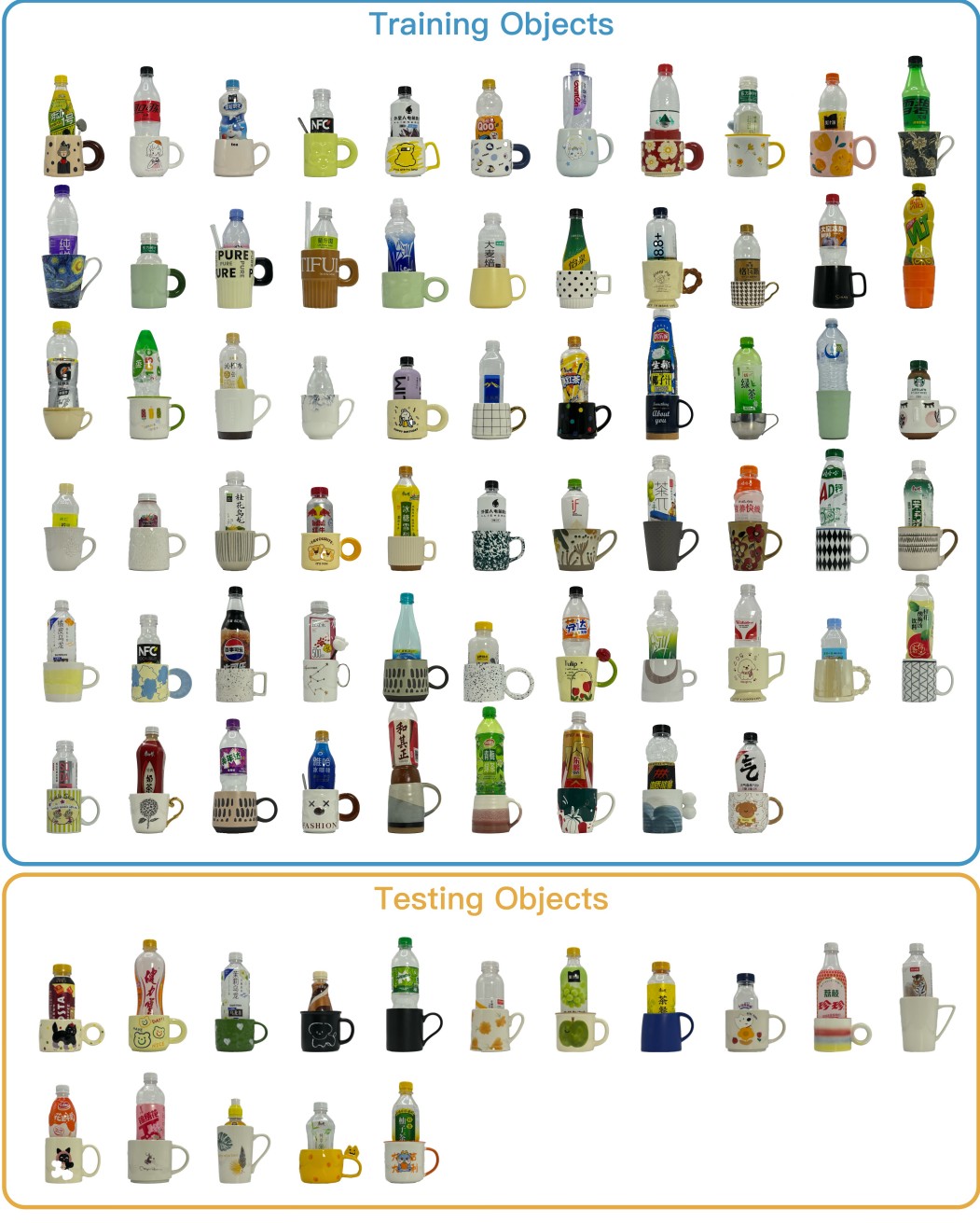

Figure 13: **Objects for Pour Water**. All of our experiments include a total of 64 training bottles and mugs, as well as 16 unseen testing bottles and mugs.

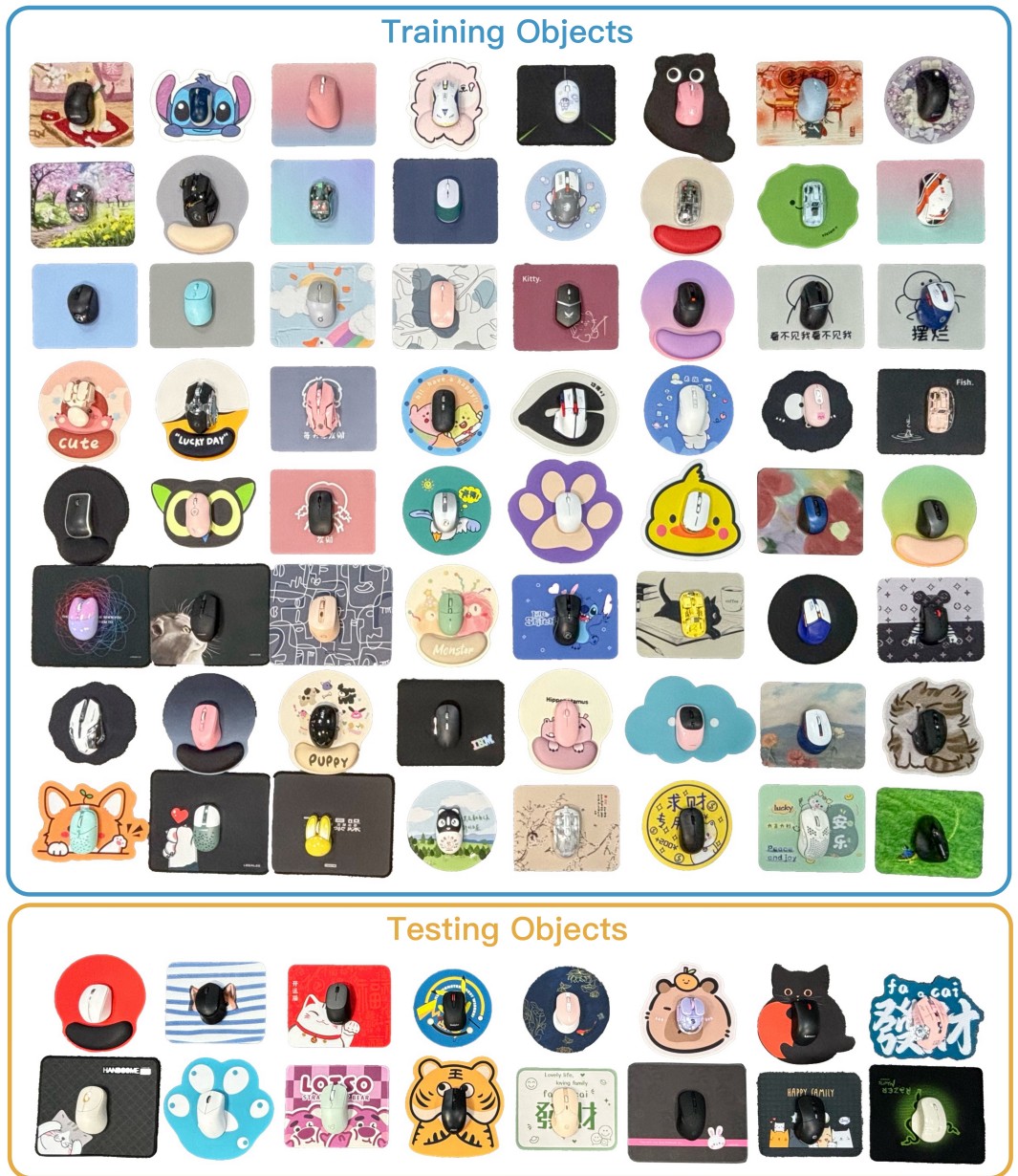

Figure 14: **Objects for `Mouse Arrangement`**. All of our experiments include a total of 64 training mice and mouse pads, as well as 16 unseen testing mice and mouse pads.

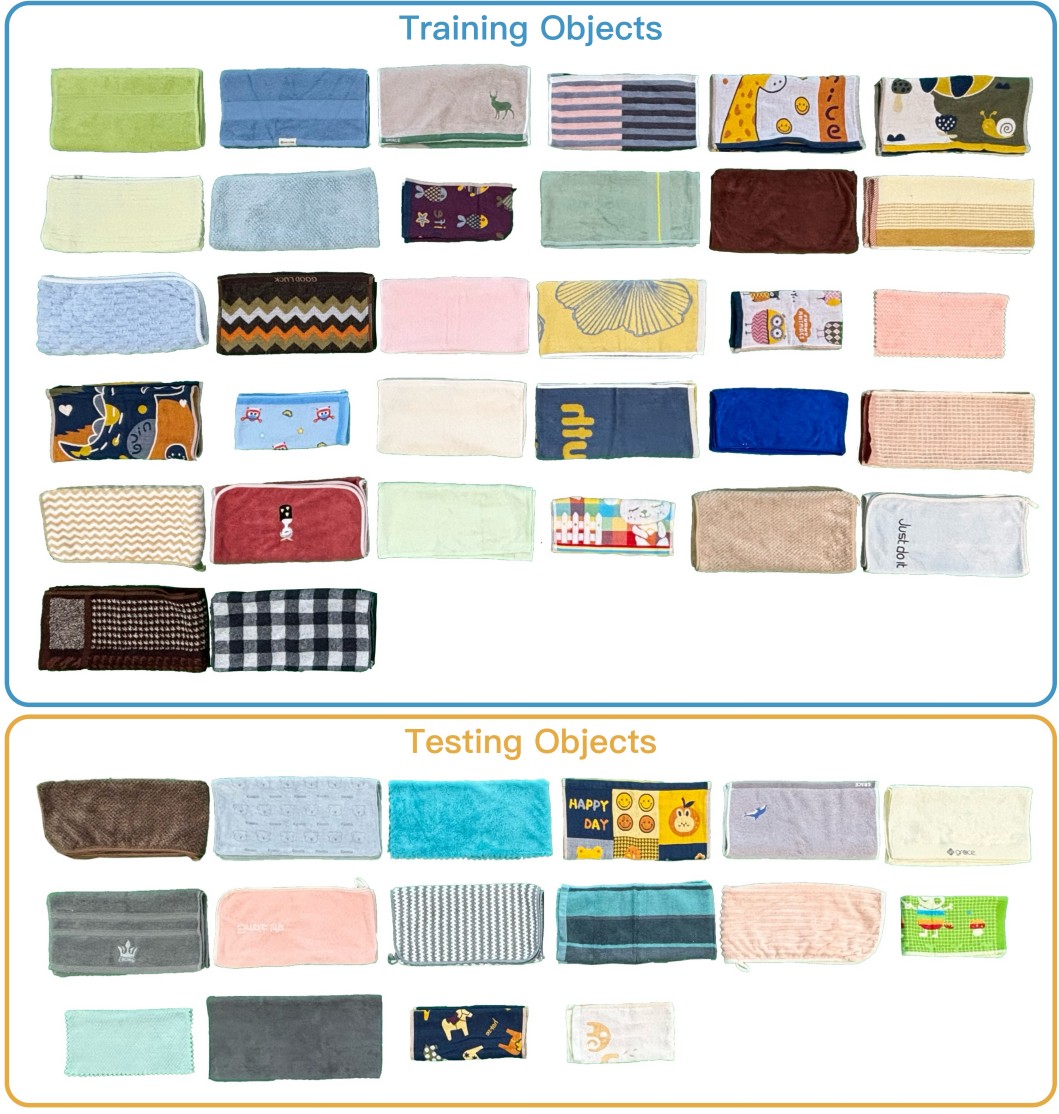

Figure 15: **Objects for `Fold Towels`**. All of our experiments include a total of 32 training towels, as well as 16 unseen testing towels.

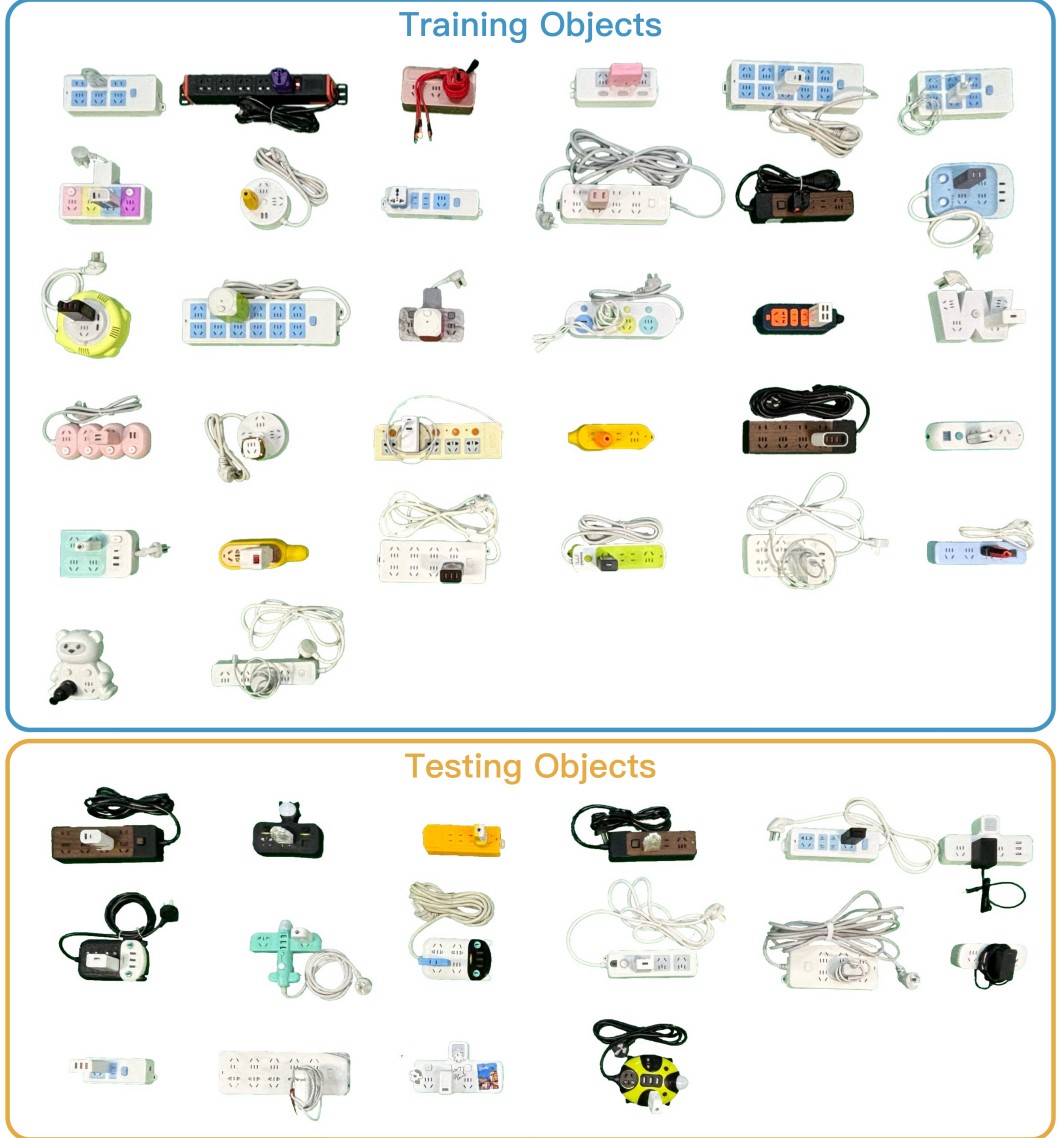

Figure 16: **Objects for `Unplug Charger`**. All of our experiments include a total of 32 training chargers and power strips, as well as 16 unseen testing chargers and power strips.

# B    DATA SOURCE

## B.1    RELATED WORK

There are three main approaches to collecting human demonstrations for robotic manipulation.

**Teleoperation.** Common teleoperation systems utilize devices such as 3D spacemouse (Zhu et al., 2020; 2023b), VR controllers (Iyer et al., 2024; Zhang et al., 2018; Cheng et al., 2024), smartphones (Mandlekar et al., 2018; Wang et al., 2023), puppeting devices (Zhao et al., 2023; Fu et al., 2024; Wu et al., 2023), or exoskeletons (Fang et al., 2023c) to allow a human operator to control a robot. This approach requires a real robot during data collection, which is expensive and limits the ability to collect large-scale data.

**Learning from human video.** This method has the potential to leverage massive Internet-scale video data (Chen et al., 2021; Bahl et al., 2022; Qin et al., 2022; Ma et al., 2022; Hu et al., 2023b; Wen et al., 2023; Zhu et al., 2024). However, these videos lack explicit action information, and there is a significant embodiment gap between humans and robots, posing substantial challenges to algorithm design.

**Hand-held grippers.** The data collected by hand-held grippers does not suffer from the embodiment gap (Song et al., 2020b; Young et al., 2021; Pari et al., 2021; Shafiullah et al., 2023; Chi et al., 2024). These devices are highly portable and intuitive to use, enabling the proactive collecting of large amounts of data and allowing for more nuanced control over the composition of the data. In this paper, we use UMI (Chi et al., 2024) as the data collection device.

## B.2    EXPERIENCE OF USING UMI TO COLLECT DATA

We share key insights gained from using UMI (Chi et al., 2024) to collect a large number of demonstrations:

**(1) Random initial pose is crucial:** For each demonstration, it's essential to randomize the initial pose of the hand-held gripper, including its height and orientation. This practice helps cover a wider range of starting conditions. Without such variation, the trained policy becomes overly sensitive to specific initial poses, limiting its effectiveness to only certain positions. Similarly, the initial position range of objects should be as extensive as possible, while remaining within the robot's kinematic and dynamic limits.

**(2) Select an environment with rich visual features:** Since UMI relies on SLAM for camera pose tracking, environments lacking sufficient visual features—such as dark areas or blank walls—can lead to tracking failures. To address this, we use the visualization tool Pangolin (Lovegrove) to verify that the environment has enough features. Introducing more distractor objects or adding textures to surfaces, such as tabletops, can both increase visual features and serve as a form of data augmentation, helping the policy learn to disregard irrelevant changes in the environment. Additionally, performing multiple mapping rounds and using batch SLAM processing can enhance the number of valid demonstrations.

**(3) Use appropriately sized manipulation objects:** Large objects that obstruct the camera's view (e.g., doors or drawers) can cause the SLAM algorithm to misinterpret the camera as stationary, leading to tracking failure. This limitation influenced our decision to avoid tasks like opening drawers, highlighting a key drawback of the current UMI. Integrating off-the-shelf pose tracking hardware (e.g., iPhone Pro or VIVE Ultimate Tracker) could potentially improve UMI's accuracy and robustness.

**(4) Additional tips:**

- Standardize behavior patterns and task completion times among different data collectors to minimize multimodal behavior in the dataset.
- When collecting data, avoid moving non-manipulation objects (distractors) and ensure that other moving entities do not enter the camera's field of view.
- Apply slight force when closing the gripper to introduce minor deformation.

## C  POLICY TRAINING

When constructing the training data, we ensure that larger datasets always contain the smaller ones. For instance, in the environment generalization experiment, if the data used to train two policies are selected from $m$ and $n$ environments (where $m < n$), the $n$ environments include all of the $m$ environments. This approach ensures a consistent data distribution across different dataset sizes, facilitating a fair comparison of policies.

To guarantee that policies trained on datasets of varying sizes can fully converge, we adjust the number of training epochs based on the total number of demonstrations. This allows policies trained on larger datasets to undergo more training steps. Specifically, the policy trained on the smallest dataset undergoes 800 epochs, totaling $5.3 \times 10^4$ training steps. The policy trained on the largest dataset undergoes 75 epochs, totaling $5 \times 10^5$ training steps, which takes 75 hours to complete on 8 A800 GPUs. We use the final checkpoint of each policy for evaluation. Given the large number of parameters in the model—the visual encoder and the noise prediction network together exceed 396 million—we use BFloat16 precision to accelerate training while maintaining numerical stability.

Our policy implementation largely follows those in Diffusion Policy (Chi et al., 2023) and UMI (Chi et al., 2024), with a minor modification: we increase the observation horizon for certain tasks. For example, in `Pour Water`, when the bottle is near the mouth of the mug, the policy initially struggles to distinguish whether it is about to start pouring or if pouring has already completed. To address this, we incorporate a more distant history step (0.25 seconds before) into the original 2-step observation horizon (which corresponds to a real-time duration of 0.05 seconds). For the `Unplug Charger` task, we incorporate a 0.5-second history step. This adjustment significantly improves the performance of `Pour Water` and `Unplug Charger` without adding much training or inference cost. For further details on the hyperparameters, refer to Table 3.

| Config | Value |
| --- | --- |
| Image observation horizon | 3 (`Pour Water`, `Unplug Charger`), 2 (other tasks) |
| Proprioception observation horizon | 3 (`Pour Water`, `Unplug Charger`), 2 (other tasks) |
| Action horizon | 16 |
| Observation resolution | 224×224 |
| Environment frequency | 5 |
| Optimizer | AdamW |
| Optimizer momentum | $\beta_1, \beta_2 = 0.95, 0.999$ |
| Learning rate for action diffusion model | 3e-4 |
| Learning rate for visual encoder | 3e-5 |
| Learning rate schedule | cosine decay |
| Batch size | 256 |
| Inference denoising iterations | 16 |
| Temporal ensemble steps | 8 |
| Temporal ensemble adaptation rate | -0.01 |

Table 3: **A default set of hyper-parameters.**

# D    TASK DETAILS

In this section, we provide a detailed introduction to four manipulation tasks and the scoring criteria during evaluation.

## D.1    POUR WATER

**Task description.** The robot performs three sequential actions: initially, it grasps a drink bottle; subsequently, it pours water into a mug; and finally, it places the bottle on a designated red coaster. The bottle is randomly placed on the table, provided it is within the robot's kinematic reach. The relative initial position of the bottle and mug is also randomized, ensuring they are spaced variably while keeping the mug visible to the camera after the bottle is grasped. The red coaster, a 9 cm diameter circle, is consistently positioned approximately 10 cm to the right of the mug and is used across all environments. This task challenges the robot's generalization capabilities due to the variability in the bottle's color, size, and height, and requires precise alignment of the bottle mouth with the mug for successful completion. The task further requires significant rotational movements, extending beyond basic pick-and-place operations. The successful execution of the pouring and placing actions critically hinges on accurately grasping the bottle initially. For testing, the bottle cap is secured tightly, and no actual water is poured out.

**Scoring criteria.**

- **Step 1: Grasping the drink bottle**
    - **0 points:** The gripper does not approach the drink bottle.
    - **1 point:** The gripper touches the drink bottle but does not grasp it due to minor errors, or it initially grasps the bottle, which then slips out during the lifting process.
    - **2 points:** The gripper pushes the drink bottle a significant distance before grasping it.
    - **3 points:** The gripper successfully grasps the drink bottle without any slippage.

- **Step 2: Pouring water into the mug**
    - **0 points:** The gripper does not approach the mug.
    - **1 point:** After rotating the drink bottle, its mouth remains outside the mug, making pouring impossible.
    - **2 points:** After rotating the drink bottle, its mouth is positioned just above the rim of the mug, allowing only partial pouring.
    - **3 points:** After rotating the drink bottle, its mouth is completely inside the mug, facilitating complete pouring.

- **Step 3: Placing the bottle on the red coaster**
    - **0 points:** The gripper does not approach the red coaster.
    - **1 point:** The drink bottle is placed outside the red coaster, or the placement process disrupts the mug, causing it to topple.
    - **2 points:** Only part of the drink bottle rests on the red coaster.
    - **3 points:** The drink bottle is fully and stably positioned on the red coaster.

**Success criteria.** A successful task requires scoring at least 2 points in Step 1, 3 points in Step 2, and at least 2 points in Step 3.

## D.2    MOUSE ARRANGEMENT

**Task description.** The robot is required to complete two steps: picking up a mouse and placing it on a mouse pad. In the first step, the mouse can be positioned anywhere on the table, as long as it remains within the robot's kinematic reach. The mouse may be oriented straight ahead, in which case the robot needs to grasp it directly from behind. Alternatively, it might be slightly tilted to the left or right, necessitating the robot to employ non-prehensile actions, such as pushing the mouse into the correct orientation before closing the gripper for picking it up. The mouse's low thickness significantly restricts the number of feasible grasping poses, leaving little margin for error, as even a slight positional deviation can cause a failed grasp. Additionally, the mouse's varying geometry

and color require the robot's policy to have strong generalization abilities, allowing it to adapt its grasping strategy based on the specific shape and size of the mouse.

**Scoring criteria.**

- **Step 1: Picking up the mouse**

    - **0 points:** The gripper does not move toward the mouse or moves around it without making contact.
    - **1 point:** The gripper approaches the correct grasping pose and touches the mouse but drops it after lifting it slightly.
    - **2 points:** The gripper pushes the mouse a significant distance before grasping it, or the mouse is grasped but falls when lifted to a higher height.
    - **3 points:** The gripper successfully grasps the mouse without any slippage.

- **Step 2: Placing the mouse on the mouse pad**

    - **0 points:** The gripper either remains stationary in the air, failing to move toward the mouse pad, or releases the mouse from a high position, causing it to fall onto the table.
    - **1 point:** The mouse is placed outside the mouse pad, or even if the entire mouse lands on the pad, it flips due to being released from a high height.
    - **2 points:** Only part of the mouse is placed on the mouse pad, or even if the entire mouse is on the pad, it bounces and shifts slightly due to being released from a relatively high height.
    - **3 points:** The gripper lowers to an appropriate height before releasing the mouse, ensuring the entire mouse is securely placed on the pad.

**Success criteria.** A successful task requires scoring 3 points in Step 1 and at least 2 points in Step 2.

### D.3 FOLD TOWELS

**Task description.** The robot is required to complete two steps: first, grasping the left edge of the towel, and second, folding the towel to the right. The initial position of the towel may vary on the table, provided it remains within the robot's kinematic reach and its tilt angle relative to the table's edge does not exceed 15 degrees. We assume the towel has already been folded several times. Manipulating deformable objects like towels presents significant challenges due to their high degrees of freedom and complex dynamics. The robot must account for the towel's softness and flexibility when selecting appropriate grasping points. Minor errors in manipulation can lead to unexpected outcomes, such as slipping or ineffective grasps. Additionally, the variety of towel styles—including differences in color, material, and texture—poses challenges for policy generalization.

**Scoring criteria.**

- **Step 1: Grasp the left edge of the towel**

    - **0 points:** The gripper does not move toward the towel or moves around it without making contact.
    - **1 point:** The gripper moves toward the towel and attempts a grasping motion but fails to grasp any towel layer.
    - **2 points:** The gripper grasps only some of the towel layers, leaving others ungrasped (since the towel has been folded multiple times, it consists of several layers).
    - **3 points:** The gripper successfully grasps all layers of the towel.

- **Step 2: Fold the towel to the right**

    - **0 points:** No folding motion toward the right is demonstrated.
    - **1 point:** After folding, the overlapping area is less than one-third of the maximum possible overlap.
    - **2 points:** After folding, the overlapping area is between one-third and two-thirds of the maximum possible overlap.

– **3 points:** After folding, the overlapping area exceeds two-thirds of the maximum possible overlap.

**Success criteria.** A successful task requires scoring 3 points in Step 1 and at least 2 points in Step 2.

### D.4 UNPLUG CHARGER

**Task description.** The robot is required to complete two steps: First, it grabs the charger that is plugged into the power strip; second, it pulls out the charger and places it on the right side of the power strip. The charger and power strip can be placed anywhere on the table as long as they remain within the robot's kinematic reach. The challenge lies in the robot's ability to accurately grasp the charger, apply sufficient force, and swiftly pull it out. Charger plugs come in different shapes and sizes, so the robot must adapt its grip to securely hold the plug.

**Scoring criteria.**

- **Step 1: Grabbing the charger**
  - **0 points:** The gripper does not grab the charger.
  - **1 point:** The gripper grabs the charger but not tightly enough, resulting in failure to pull out the charger.
  - **2 points:** The gripper securely holds the charger, but during the process, there is a collision with the power strip, though the charger is eventually pulled out.
  - **3 points:** The gripper securely holds the charger without colliding with other objects, and the charger is successfully pulled out afterward.

- **Step 2: Pulling out the charger**
  - **0 points:** The charger is not pulled out.
  - **2 points:** After pulling out the charger, it slips from the gripper.
  - **3 points:** The charger is successfully pulled out, and the gripper places it to the right side of the power strip.

**Success criteria.** A successful task requires scoring at least 2 points in Step 1 and 3 points in Step 2.

# E EVALUATION

## E.1 COMPARISON OF EVALUATION METRICS

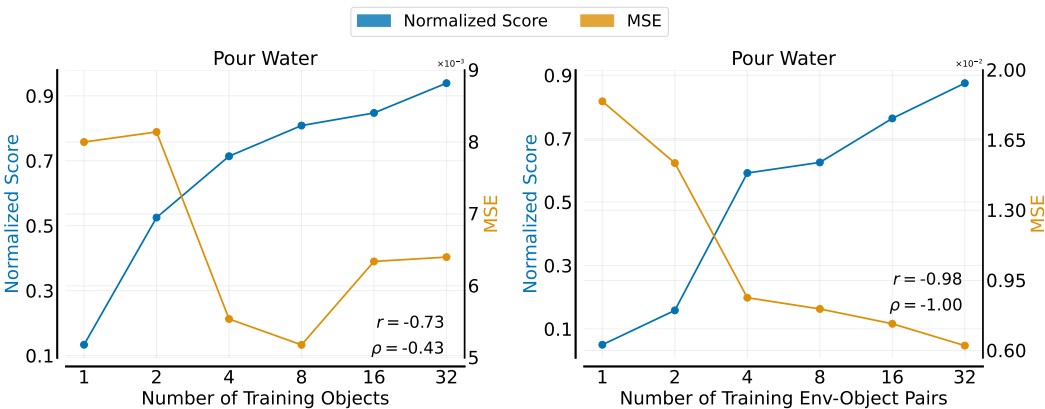

Figure 17: **Comparison between normalized score and MSE. Left:** In the object generalization experiment, the inverse correlation between MSE and normalized score is weak. **Right:** In the generalization experiment across both environments and objects, the inverse correlation between MSE and normalized score is very strong. Correlation coefficients (Pearson's $r$ and Spearman's $\rho$) are shown in the bottom right.

We use tester-assigned scores as our primary evaluation metric, acknowledging that this approach inherently introduces some subjectivity from the testers. An alternative metric, the mean squared error (MSE) on the validation set, offers a potential objective measure that does not require human intervention. In this section, we provide a detailed comparison of these two metrics. To calculate MSE, we collect 30 human demonstrations for each evaluation environment or object, forming the validation set. We then compute the MSE by averaging the squared differences between the policy-predicted actions and the human actions at each timestep.

We observe a strong inverse correlation between MSE and normalized scores in certain cases. For example, in the right plot of Figure 17, the experimental setup evaluates the policy's generalization across both environments and objects on `Pour Water`. As the number of training environment-object pairs increases, the normalized score gradually rises while the MSE steadily decreases, with Pearson's $r = -0.98$ and Spearman's $\rho = -1.00$. This suggests that MSE could potentially replace the human scoring method. However, in certain scenarios, MSE does not correlate well with real-world performance. For example, in the left plot of Figure 17, the experiment evaluates the policy's generalization across objects on `Pour Water`. When the number of training objects increases to 16, the MSE actually increases, resulting in a Pearson's $r$ of only $-0.73$. Similarly, in experiments exploring model training strategies (Section 6), the MSE for LoRA is significantly lower than for full fine-tuning (0.0049 vs. 0.006). Nevertheless, in real-world tests, LoRA's policy performs worse than full fine-tuning, with normalized scores of 0.72 and 0.9, respectively.

Overall, the MSE on the validation set often does not correlate with real-world performance, and many anomalies appear unpredictably and without discernible patterns. This leads us to believe that MSE is not a completely reliable evaluation metric. In practice, we use MSE more as a debugging tool to quickly identify policies with obvious problems.

## E.2 EVALUATION WORKFLOW

We use the environment generalization experiment as a case study to demonstrate our evaluation workflow. Recall that in this experiment, we collect data with the same object across 32 environments, training a total of 21 policies. Each policy is evaluated in 8 unseen environments using the same object, with 5 trials per environment. The average normalized score from these 40 trials is reported for each policy. Operationally, we complete the training of all 21 policies before deploying the entire robotic system (refer to Appendix F) into a new environment for evaluation. To ensure

unbiased results, we conduct blind tests: initially, we set an initial position for the object and randomly shuffle the order of the 21 policies. Testing then proceeds at this initial position, with the 21 shuffled policies scored according to the criteria outlined in Appendix D. Subsequently, we select a new initial position for the object and repeat the scoring process for the 21 shuffled policies. This procedure is replicated five times to conclude the testing in one environment. The robot system is then transitioned to another new environment, and the entire process is repeated, completing eight cycles in total for all tests.

Such an evaluation workflow ensures that policies evaluated within the same batch can be directly compared—since they are exposed to the same conditions—but comparisons across different batches are not valid. This restriction arises because the environments and the initial object positions can vary between batches. For instance, despite the data in Fig. 4 and Fig. 6 being evaluated under the same conditions—across eight unseen environments and using two unseen objects per environment, each with five trials—they cannot be directly compared since they originate from separate batches.

## F    HARDWARE SETUP

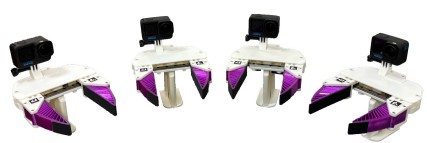

Figure 18: **UMI hand-held grippers.** We do not install side mirrors on the grippers.

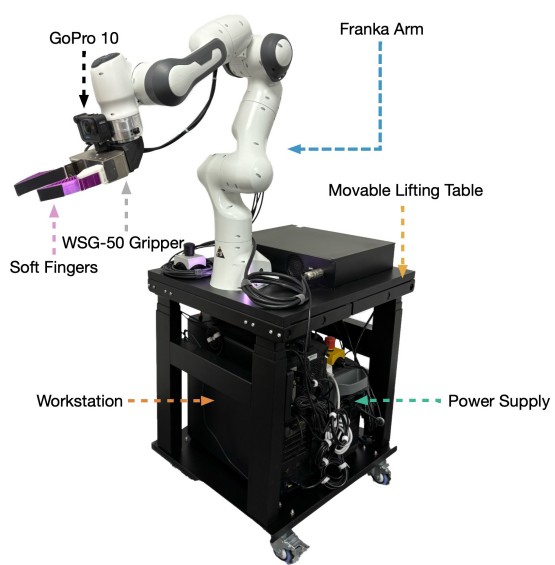

Figure 19: **Deployment hardware setup.**

The comprehensive hardware building guide for the hand-held gripper can be found at: `https://umi-gripper.github.io/`. Figure 18 displays the four hand-held grippers used in our study. Next, we introduce our deployment hardware setup, shown in Figure 19. We use a Franka Emika Panda robot (a 7-DoF arm) equipped with a Weiss WSG-50 gripper (a 1-DoF parallel jaw gripper). To address the robot's limited end-effector pitch, we utilize a mounting adapter designed by Chi et al. (2024) to rotate the WSG-50 gripper by 90 degrees relative to the robot's end-effector flange. The gripper is equipped with soft, compliant fingers printed using purple 95A TPU material. For perception, we use a wrist-mounted GoPro Hero 10 camera with a fisheye lens. Real-time video streaming from the GoPro is achieved through a combination of the GoPro Media Mod and the Elgato HD60 X external capture card. Policy inference is performed on a workstation equipped with an NVIDIA 4090 GPU (24 GB VRAM). All components are powered by a mobile power supply (EcoFlow DELTA 2 Max) with a 2048 Wh capacity, which also serves as a 23 kg counterweight to prevent tipping. The system is mounted on a custom movable lifting table. While the table cannot move autonomously, its mobility allows for testing our policies in non-laboratory settings.

# G ADDITIONAL EXPERIMENTAL RESULTS

## G.1 DATA SCALING LAWS ON MSE

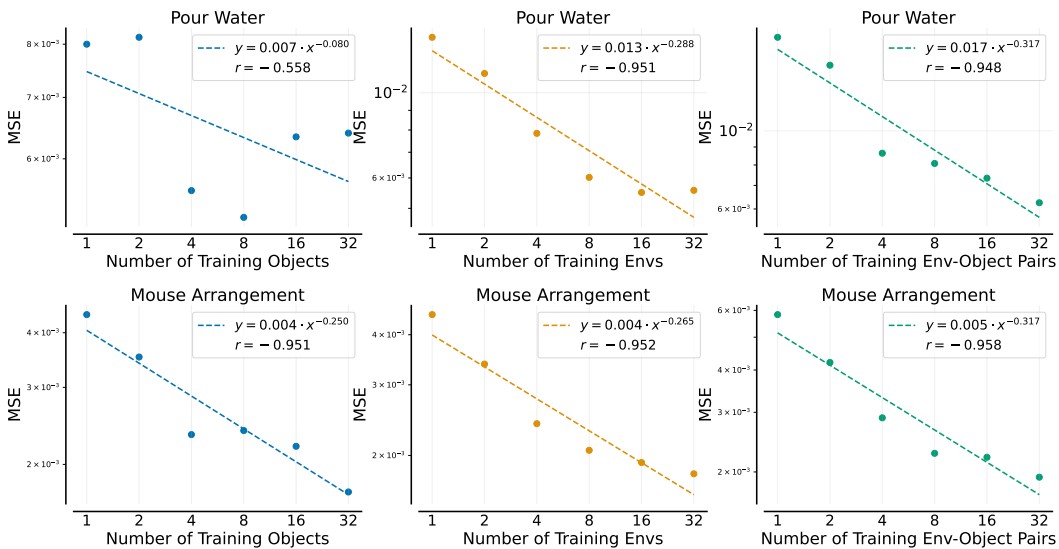

Figure 20: **Data scaling laws on MSE.** Dashed lines represent power-law fits, with the equations provided in the legend. All axes are shown on a logarithmic scale.

In the main text, we present power-law data scaling laws based on tester-assigned scores (Section 4.2). In this section, inspired by the scaling laws on cross-entropy loss observed in large language models (Kaplan et al., 2020), we explore whether our power-law relationships also hold for action mean squared error (MSE). To calculate the MSE, we collect 30 human demonstrations for each evaluation environment or object, forming a validation set. We then compute the MSE by averaging the squared differences between the predicted actions and the human actions at each timestep.

Similar to Fig. 5, we fit a linear model to the log-transformed data and present the results in Fig. 20. As shown in Fig. 20, in most cases the absolute value of the correlation coefficient $r$ is relatively large, indicating that power-law data scaling laws generally hold for MSE as well. However, compared to Fig. 5, we observe that all absolute values of $r$ in Fig. 20 are smaller, suggesting a weaker scaling trend for MSE. Notably, in certain cases—such as the second column of Fig. 20 (in `Pour Water`)—the absolute value of $r$ is unusually low (only 0.558), primarily due to outliers in the MSE. Such abnormal MSE values are not uncommon, as we discuss in detail in Appendix E.1.

While MSE can serve as a reasonable proxy metric when time-consuming human evaluations are not feasible, it cannot fully capture the true performance of a closed-loop visuomotor policy in the real world. We believe that tester-assigned scores better reflect the policy's actual performance. Therefore, we choose them as the primary evaluation metric and present the data scaling laws based on this metric in the main text.

## G.2 KEEPING THE TOTAL NUMBER OF DEMONSTRATIONS CONSTANT

In the main text, we observe that as the number of training objects, environments, or environment-object pairs increases, the policy's generalization ability improves. However, this increase is accompanied by a rise in the total number of demonstrations. In Figures 21, 22, and 23, we redraw the plots to maintain a relatively constant total number of demonstrations while varying the number of objects, environments, or environment-object pairs. This adjustment does not require rerunning the experiments; instead, we connect points from the original plots that correspond to a similar number of demonstrations. Although the points along the same line do not represent exactly the same number of demonstrations, they are approximately equivalent. For instance, "2×" denotes around 200 demonstrations. We exclude the "1×" line (approximately 100 demonstrations) because the results become unstable and unreliable at that level when dealing with larger numbers of environments or

objects. From these plots, we see that even when controlling for the total number of demonstrations, increasing the number of environments or objects still enhances the policy's generalization performance, as seen most clearly in Figure 23. Additionally, the total number of demonstrations required for the policy's performance to saturate appears moderate. Once the total reaches around $16\times$ (approximately 1600 demonstrations), further increases offer minimal performance improvements.

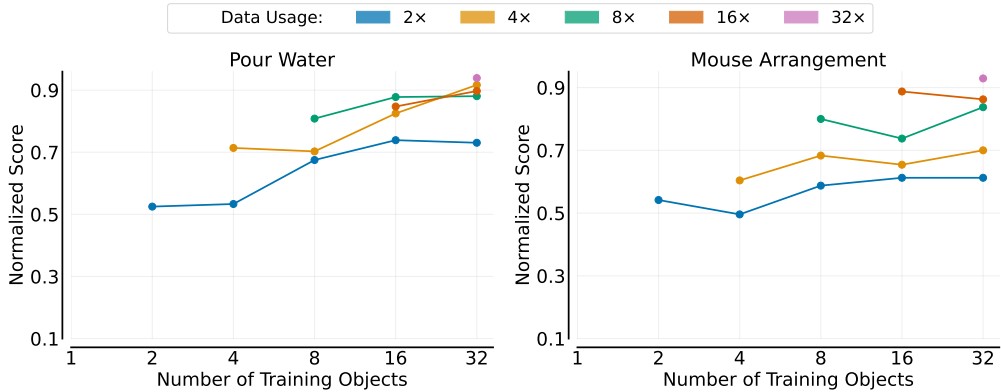

Figure 21: **Object generalization.** Each curve corresponds to a different total numbers of demonstrations used, with normalized scores shown as a function of the number of training objects.

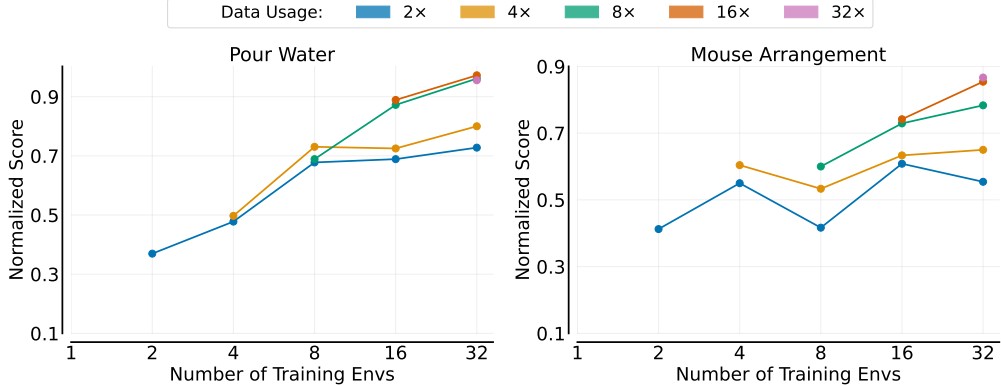

Figure 22: **Environment generalization.** Each curve corresponds to a different total numbers of demonstrations used, with normalized scores shown as a function of the number of training environments.

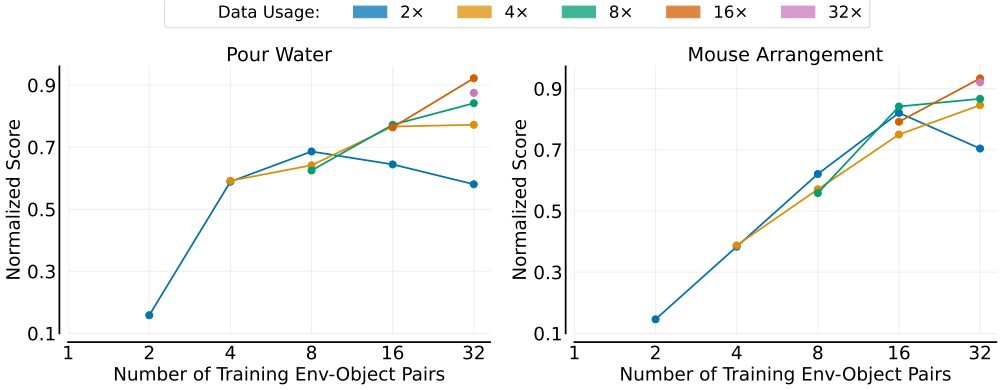

Figure 23: **Generalization across environments and objects.** Each curve corresponds to a different total numbers of demonstrations used, with normalized scores shown as a function of the number of training environment-object pairs.

## G.3 RAW TEST SCORES

In this section, we present the raw test scores before normalization. The scores for `Pour Water` are shown in Table 4, Table 5, Table 6, and Table 7. The scores for `Mouse Arrangement` are shown in Table 8, Table 9, Table 10, and Table 11.

| #Objs \ Usage | 3.125% | 6.25% | 12.5% | 25% | 50% | 100% |
|---|---|---|---|---|---|---|
| 1 | | | | | | 1.2 |
| 2 | | | | | 3.175 | 4.725 |
| 4 | | | | 4.55 | 4.8 | 6.425 |
| 8 | | | 4.575 | 6.075 | 6.325 | 7.275 |
| 16 | | 3.6 | 6.65 | 7.425 | 7.9 | 7.625 |
| 32 | 2.45 | 6.575 | 8.25 | 7.925 | 8.075 | 8.45 |

Table 4: **Object generalization on `Pour Water`**. Normalizing these scores by dividing them by 9 yields the results shown in Fig. 2.

| #Envs \ Usage | 3.125% | 6.25% | 12.5% | 25% | 50% | 100% |
|---|---|---|---|---|---|---|
| 1 | | | | | | 1.3 |
| 2 | | | | | 2.85 | 3.325 |
| 4 | | | | 2.55 | 4.3 | 4.475 |
| 8 | | | 3.925 | 6.1 | 6.575 | 6.2 |
| 16 | | 4.15 | 6.2 | 6.525 | 7.85 | 8 |
| 32 | 3.475 | 6.55 | 7.2 | 8.65 | 8.75 | 8.6 |

Table 5: **Environment generalization on `Pour Water`**. Normalizing these scores by dividing them by 9 yields the results shown in Fig. 3.

| #Pairs \ Usage | 3.125% | 6.25% | 12.5% | 25% | 50% | 100% |
|---|---|---|---|---|---|---|
| 1 | | | | | | 0.45 |
| 2 | | | | | 1.65 | 1.425 |
| 4 | | | | 2.725 | 5.3 | 5.325 |
| 8 | | | 4.95 | 6.175 | 5.775 | 5.625 |
| 16 | | 4.8 | 5.8 | 6.9 | 6.95 | 6.875 |
| 32 | 3.95 | 5.225 | 6.95 | 7.575 | 8.3 | 7.875 |

Table 6: **Generlization across environments and objects on `Pour Water`**. Normalizing these scores by dividing them by 9 yields the results shown in Fig. 4.

| #Demos | 64 | 100 | 200 | 400 | 800 | 1600 | 3200 | 6400 |
|---|---|---|---|---|---|---|---|---|
| Score | 4.35 | 6.15 | 6.875 | 7.025 | 6.975 | 7.2 | 7.125 | 6.525 |

Table 7: **Number of demonstrations on `Pour Water`**. Normalizing these scores by dividing them by 9 yields the results shown in Fig. 7.

| Usage #Objs | 3.125% | 6.25% | 12.5% | 25% | 50% | 100% |
|---|---|---|---|---|---|---|
| 1 | | | | | | 1.3 |
| 2 | | | | | 2.475 | 3.25 |
| 4 | | | | 2.425 | 2.975 | 3.625 |
| 8 | | | 1.75 | 3.525 | 4.1 | 4.8 |
| 16 | | 2.525 | 3.675 | 3.925 | 4.425 | 5.325 |
| 32 | 3.7 | 3.675 | 4.2 | 5.025 | 5.175 | 5.575 |

Table 8: **Object generalization on `Mouse Arrangement`**. Normalizing these scores by dividing them by 6 yields the results shown in Fig. 2.

| Usage #Envs | 3.125% | 6.25% | 12.5% | 25% | 50% | 100% |
|---|---|---|---|---|---|---|
| 1 | | | | | | 1.3 |
| 2 | | | | | 1.975 | 2.475 |
| 4 | | | | 1.8 | 3.3 | 3.625 |
| 8 | | | 2.075 | 2.5 | 3.2 | 3.6 |
| 16 | | 1.525 | 3.65 | 3.8 | 4.375 | 4.45 |
| 32 | 2.725 | 3.325 | 3.9 | 4.7 | 5.125 | 5.2 |

Table 9: **Environment generalization on `Mouse Arrangement`**. Normalizing these scores by dividing them by 6 yields the results shown in Fig. 3.

| Usage #Pairs | 3.125% | 6.25% | 12.5% | 25% | 50% | 100% |
|---|---|---|---|---|---|---|
| 1 | | | | | | 0.75 |
| 2 | | | | | 0.975 | 0.875 |
| 4 | | | | 1.8 | 2.3 | 2.325 |
| 8 | | | 2.425 | 3.725 | 3.425 | 3.35 |
| 16 | | 3.375 | 4.925 | 4.5 | 5.05 | 4.75 |
| 32 | 4.225 | 4.225 | 5.075 | 5.2 | 5.6 | 5.525 |

Table 10: **Generlization across environments and objects on `Mouse Arrangement`**. Normalizing these scores by dividing them by 6 yields the results shown in Fig. 4.

| #Demos | 64 | 100 | 200 | 400 | 800 | 1600 | 3200 | 6400 |
|---|---|---|---|---|---|---|---|---|
| Score | 1.725 | 3.025 | 3.3 | 3.775 | 3.975 | 3.8 | 3.875 | 3.8 |

Table 11: **Number of demonstrations on `Mouse Arrangement`**. Normalizing these scores by dividing them by 6 yields the results shown in Fig. 7.

## G.4   SUCCESS RATE

Table 12 presents the success rates of the policy trained across 32 environment-object pairs for each task. The detailed criteria for task success are provided in Appendix D.

| Task | Environment ID | | | | | | | | Mean |
|---|---|---|---|---|---|---|---|---|---|
| | 1 | 2 | 3 | 4 | 5 | 6 | 7 | 8 | |
| Pour Water | 80% | 40% | 100% | 80% | 100% | 100% | 80% | 100% | 85% |
| Mouse Arrangement | 100% | 80% | 100% | 100% | 80% | 80% | 100% | 100% | 92.5% |
| Fold Towels | 100% | 100% | 60% | 100% | 100% | 60% | 100% | 80% | 87.5% |
| Unplug Charger | 80% | 60% | 100% | 100% | 100% | 80% | 100% | 100% | 90% |

Table 12: **Success rate across all tasks.** For each task, we report the success rate in each evaluation environment.

