# OpenReview forum: "Data Scaling Laws in Imitation Learning for Robotic Manipulation"
_ICLR.cc/2025/Conference — ICLR 2025 Oral_

### Official Review · Reviewer_SnY8 · 2024-10-26

**Soundness:** 3
**Presentation:** 4
**Contribution:** 4
**Rating:** 8
**Confidence:** 4

**Summary:**

In this paper, the authors seek to establish data scaling laws for the generalisation performance of robot manipulation policies trained on individual tasks where the policy is learnt via the imitation learning. The authors focus on establishing data scaling laws with respect to two features of the training data, the number of unique objects from an individual object category (e.g. bottles) and the number of unique environments. Through collecting demonstration data with handheld grippers (UMI) and evaluating policies trained on this data using a well-defined human scoring rubric, the authors establish a power law relationship between their scoring rubric and the policy generalisation performance. In this case, generalisation performance is examined only with respect to unseen objects and environments and the policy architecture reflect current SOTA  through leveraging diffusion policies. This paper provides important insights into data scaling laws in robotics and contributes valuable discussion to understanding effective data collection strategies for learning robot manipulation policies via imitation learning.

**Strengths:**

- The authors effectively contribute to the discussion of scaling laws for robot learning through extensive experiments in generalisation across unseen objects and environments.

- The authors provide practical validation of their findings through using their insights to establish optimal data collection strategies for training policies, which they demonstrate on another set of tasks.

- The authors provide a useful discussion around evaluation criteria for task success and provide an example of defining a scoring rubric for evaluating task success on specific tasks.

- The authors provide ample information on their experimental setup and ensure that their experiments are reproducible.

- The overall quality of the experiments and discussion is good, while there are some limitations to this work the authors address many of these limitations in their discussions and provide solid experimental evidence for their claims. This is a valuable contribution to scaling laws for robot learning, while more work needs to be done overall I find this paper to be valuable.

**Weaknesses:**

- Generalisation performance is only discussed for policy learning in the single-task setting, the paper doesn't consider the multi-task setting which remains very relevant in modern robot learning applications.

- The scale of the demonstration datasets is limited, more work is required to ensure that the scaling laws hold true at larger scales.

- Generalisation performance is only discussed with respect to two criteria, the authors also argue that these criteria encapsulates all factors a policy may encounter in real-world deployment. In general, I would say that this argument is incorrect.

"We categorize generalization into two dimensions: environment generalization and object
generalization, which essentially encompass all factors a policy may encounter during real-world
deployment."

- The authors argue that they focus on out-of-domain generalisation, however they do not quantify out-of-domain in their experiments. This is more generally a challenge for evaluating generalisation performance of machine learning experiments. It remains a weakness of the current approach as we aren't quantifying out-of-domain which makes it challenging to comment on generalisation performance. With this being said the appendix provides visuals of the objects and environments which provides sufficient evidence for data diversity to support the authors claims.

- The scoring rubric for tasks has the potential to introduce subjective bias. In spite of this potential weakness, the authors effectively argue the need for this rubric versus more rudimentary metrics such as MSE. The results of this work hinge on a readers confidence in the scoring rubric for experiments, in general I feel confident in the quality of this work, however, it is important to mention this point.

**Questions:**

1. I believe the following argument is flawed and would like to hear your thoughts on this point:

"We categorize generalization into two dimensions: environment generalization and object
generalization, which essentially encompass all factors a policy may encounter during real-world
deployment."

Otherwise your paper was unambiguous so I have no further questions right now, congratulations on this work and thank you for contributing it to the robot learning community.

---

> ### Author Response · Authors · 2024-11-17
>
> We thank Reviewer SnY8 for their insightful and positive review.
>
> > Generalisation performance is only discussed for policy learning in the single-task setting, the paper doesn't consider the multi-task setting which remains very relevant in modern robot learning applications.
>
> Yes, as we discussed in the limitations section of the paper, we do not consider task-level generalization at this stage. We believe that addressing this would require collecting vast amounts of data from thousands of tasks, which is beyond the scope of our current work. We hope to explore task-level generalization in the future when more resources become available.
>
> > The scale of the demonstration datasets is limited, more work is required to ensure that the scaling laws hold true at larger scales.
>
> Thank you for bringing this up. Compared to the data used in current large language models, our dataset is indeed relatively small. In our future work, we will verify our conclusions on larger datasets and a more complex set of tasks.
>
> > Generalisation performance is only discussed with respect to two criteria, the authors also argue that these criteria encapsulates all factors a policy may encounter in real-world deployment. In general, I would say that this argument is incorrect.
>
> Thank you for this insightful suggestion. Our previous statement may indeed have been too absolute. Our intention was to convey that for most single tasks (excluding task-level generalization for now), an ideal robot policy deployed in the real world should handle new environments and objects. Our paper primarily studies environments and object generalization, but there are many other factors we have not considered—for example, motion-level generalization is needed when the environment contains obstacles. If humans are present, we need to consider the safety of interacting with them and the robustness to various human disturbances, and so on. We also look forward to hearing your thoughts on generalization.
> > The authors argue that they focus on out-of-domain generalisation, however they do not quantify out-of-domain in their experiments. This is more generally a challenge for evaluating generalisation performance of machine learning experiments. It remains a weakness of the current approach as we aren't quantifying out-of-domain which makes it challenging to comment on generalisation performance. With this being said the appendix provides visuals of the objects and environments which provides sufficient evidence for data diversity to support the authors claims.
>
> Thank you for pointing this out. We fully agree with your perspective that quantifying out-of-domain generalization is a challenge faced by all machine learning experiments, especially in in-the-wild experiments like ours. Therefore, we define out-of-domain as the policy’s ability to work on unseen instances in both the environment and object dimensions. Here, “unseen” does not refer to only changing partial layouts or settings of the environment, or partial attributes of the objects, but rather to working in entirely new environments (e.g., a new restaurant) and with new objects (e.g., a newly purchased cup). As you have observed, we provided images of the environments and objects in the appendix to support this point.
>
> > The scoring rubric for tasks has the potential to introduce subjective bias. In spite of this potential weakness, the authors effectively argue the need for this rubric versus more rudimentary metrics such as MSE. The results of this work hinge on a readers confidence in the scoring rubric for experiments, in general I feel confident in the quality of this work, however, it is important to mention this point.
>
> Thank you for this important observation. The evaluation metric is indeed another challenge in robot learning. We believe that the scoring rubric captures more nuanced behaviors and reflects the robot’s real performance, but it inevitably introduces subjective bias. We will make this point clearer in the final version of the paper.

---

> > ### Comment · Reviewer_SnY8 · 2024-11-26
> > **Final Response**
> >
> > "Yes, as we discussed in the limitations section of the paper, we do not consider task-level generalization at this stage. We believe that addressing this would require collecting vast amounts of data from thousands of tasks, which is beyond the scope of our current work. We hope to explore task-level generalization in the future when more resources become available."
> >
> > This is completely understandable from my point of view, I think the level of work required to collate the current set of results is significant let alone tackling the multi-task setting, nonetheless I felt this was relevant to include in my review.
> >
> > "Thank you for bringing this up. Compared to the data used in current large language models, our dataset is indeed relatively small. In our future work, we will verify our conclusions on larger datasets and a more complex set of tasks."
> >
> > Thanks for clarifying and looking forward to follow up work in this direction.
> >
> > "Thank you for this insightful suggestion. Our previous statement may indeed have been too absolute. Our intention was to convey that for most single tasks (excluding task-level generalization for now), an ideal robot policy deployed in the real world should handle new environments and objects. Our paper primarily studies environments and object generalization, but there are many other factors we have not considered—for example, motion-level generalization is needed when the environment contains obstacles. If humans are present, we need to consider the safety of interacting with them and the robustness to various human disturbances, and so on. We also look forward to hearing your thoughts on generalization."
> >
> > Thank you for acknowledging this, I believe in the context of the paper this is not a major issue, however, the referenced statement in the paper did feel too absolute so I thought to mention it in my review.
> >
> > "Thank you for pointing this out. We fully agree with your perspective that quantifying out-of-domain generalization is a challenge faced by all machine learning experiments, especially in in-the-wild experiments like ours. Therefore, we define out-of-domain as the policy’s ability to work on unseen instances in both the environment and object dimensions. Here, “unseen” does not refer to only changing partial layouts or settings of the environment, or partial attributes of the objects, but rather to working in entirely new environments (e.g., a new restaurant) and with new objects (e.g., a newly purchased cup). As you have observed, we provided images of the environments and objects in the appendix to support this point."
> >
> > Thanks for the response on this point.
> >
> > "Thank you for this important observation. The evaluation metric is indeed another challenge in robot learning. We believe that the scoring rubric captures more nuanced behaviours and reflects the robot’s real performance, but it inevitably introduces subjective bias. We will make this point clearer in the final version of the paper."
> >
> > Thank you and once again congratulations on this work. It is a valuable contribution in my opinion and raises some important questions that can be addressed in future work.

---

### Official Review · Reviewer_facp · 2024-10-29

**Soundness:** 3
**Presentation:** 4
**Contribution:** 3
**Rating:** 8
**Confidence:** 4

**Summary:**

This paper presents a comprehensive analysis of the robot's generalization ability to novel objects and environments related to data scaling. This identifies two main findings: (i) generalization generally follows an approximate power-law with respect to the number of of environments and objects, and (ii) increasing the diversity of objects and environments enhances performance more effectively than merely gathering more data in limited setups.

**Strengths:**

- This study is follows a rigorous protocol to examine scaling laws across diverse environments and objects, with a fair evaluation method to minimize evaluation bias.
- The paper presents efficient data collection strategies, which are crucial for the robot learning field, where data collection is costly and limited.
- This work plans to open-source the code, data, and model, which can be useful for future research on robot learning generalization.

**Weaknesses:**

- **Details on data**
  + Considering the multiple other factors that can affect learning performance, the authors should provide data details for each environment, object, and environment-object variation. This could include specifying the number of demonstrators [1] (with their IDs), listing the data collection protocol [2], and showing relevant statistics (e.g., the number of failed demos, action variance [3], and task horizon for each demonstrator [3]). Such information would clarify that any observed variation is due solely to the environment, object, or environment-object pair rather than bias in the data itself.
  + One concern about particular data collection hardware (i.e., UMI) is that it may introduce added bias. For instance, if a specific teleoperator is highly skilled, they might gather higher-quality data by using optimal speed movements or demonstrating behaviors at the precise height where the robot is positioned. The authors could clarify this in relation to the above point.
- **Limited algorithmic variations**
  + The experiments use only a single algorithm with a single seed, which may restrict to make a comprehensive conclusion.

### Referneces
- [1] Mandlekar et al., "What Matters in Learning from Offline Human Demonstrations for Robot Manipulation"
- [2] Zhao et al., "ALOHA Unleashed: A Simple Recipe for Robot Dexterity"
- [3] Belkhale et al., "Data Quality in Imitation Learning"

**Questions:**

- Figure 5 seems to lack an explanation of how to obtain $\alpha$ and $\beta$.
- Does each item in the heatmap in Figure 6 represent the same number of demonstrations? The relevant context was difficult to locate in the paper.
- Including an in-domain evaluation would help readers better understand the performance gap between zero-shot evaluation and in-domain data.
- Supplementary suggestion: It would be interesting to explore if the scaling law applies to environments/objects generated by the generative model [1]. Investigating this could reveal a potential gap between synthesized and real-world data and help identify the factors influencing it.

[1] Yu et al., "Scaling Robot Learning with Semantically Imagined Experience"

---

> ### Author Response · Authors · 2024-11-17
>
> We thank Reviewer facp for their thorough and insightful review.
>
> > Considering the multiple other factors that can affect learning performance, the authors should provide data details for each environment, object, and environment-object variation. This could include specifying the number of demonstrators [1] (with their IDs), listing the data collection protocol [2], and showing relevant statistics (e.g., the number of failed demos, action variance [3], and task horizon for each demonstrator [3]). Such information would clarify that any observed variation is due solely to the environment, object, or environment-object pair rather than bias in the data itself.
>
> Thank you for your insightful suggestion. In the table below, we have listed statistics of some data. Specifically, we present the data for the tasks of **Pour Water** and **Mouse Arrangement**, which were used in the experiments on generalization across both environments and objects, and for the tasks of **Fold Towels** and **Unplug Charger**, which were used when verifying the data collection strategy. The table includes the total number of valid demonstrations (after SLAM filtering), the number of valid demonstrations collected by each collector (with their IDs), and the average time horizon for each task.
>
> | Task              | Total Valid Demos | **Demos per Collector** | Time Horizon (s) |
> | :---------------- | :---------------- | :---------------------- | :--------------- |
> | Pour Water        | 3648              | id_1: 1513              | 5.13 ± 0.64      |
> |                   |                   | id_2: 959               |                  |
> |                   |                   | id_3: 1176              |                  |
> | Mouse Arrangement | 3564              | id_1: 871               | 3.93 ± 0.85      |
> |                   |                   | id_2: 885               |                  |
> |                   |                   | id_3: 909               |                  |
> |                   |                   | id_4: 899               |                  |
> | Fold Towels       | 1752              | id_1: 473               | 3.26 ± 0.48      |
> |                   |                   | id_2: 483               |                  |
> |                   |                   | id_3: 451               |                  |
> |                   |                   | id_4: 345               |                  |
> | Unplug Charger    | 1733              | id_1: 483               | 3.63 ± 0.49      |
> |                   |                   | id_2: 507               |                  |
> |                   |                   | id_3: 444               |                  |
> |                   |                   | id_4: 299               |                  |
>
> We have provided the protocol instructions given to data collectors here: [Data Collection Protocol](https://anonymous-data-scaling.github.io/media/Data_Collection_Protocol.pdf). We will release all data, code, and models associated with our work. We hope this information helps clarify any concerns regarding data variations.
>
> > One concern about particular data collection hardware (i.e., UMI) is that it may introduce added bias. For instance, if a specific teleoperator is highly skilled, they might gather higher-quality data by using optimal speed movements or demonstrating behaviors at the precise height where the robot is positioned. The authors could clarify this in relation to the above point.
>
> Thank you for bringing this to our attention. We have also noticed that data quality significantly impacts the policy’s performance. Therefore, we ensured that our data is of high quality and that the behaviors are consistent. Before collecting the final data used in our experiments, we made sure that all four data collectors were highly skilled, and we provided detailed instructions for each task being performed (see our previous response). Ensuring high-quality and consistent data is manageable with only four data collectors. However, we recognize that scaling up to a larger number—say, 100 data collectors—would make it challenging to guarantee the same level of data quality and consistency. Exploring how data of varying quality affects the policy and how to learn from such highly multimodal data is a very important issue, which we leave for future work.

---

> > ### Author Response · Authors · 2024-11-17
> >
> > > The experiments use only a single algorithm with a single seed, which may restrict to make a comprehensive conclusion.
> >
> > Thank you for pointing this out. This is indeed a limitation of our current work. However, it is worth noting that Diffusion Policy is the current state-of-the-art imitation learning algorithm and is highly representative. Additionally, all our experiments are executed on a real robot arm in reasonably cluttered environments. Conducting experiments with multiple seeds would mean a multiplied, unaffordable amount of experimental time in the real world. Moreover, it is worth noting that the different policies presented in the same figures or tables were initialized with different seeds, which actually provides sufficient diversity. This variation helps to partially address concerns about the robustness of our results. We look forward to exploring more algorithms in the future—such as ACT—to see how they affect data scaling laws. We also anticipate establishing a diverse simulation environment to support testing generalization and conducting experiments with multiple seeds.
> >
> > > Figure 5 seems to lack an explanation of how to obtain $\alpha$ and $\beta$.
> >
> > To determine whether two variables  $Y$  and  $X$  satisfy a power-law relationship, we can transform the original data by taking the logarithm and fitting a simple linear regression model to the transformed data  $\log(Y)$  and  $\log(X)$ . The slope of the regression line provides the exponent  $\alpha$ , and the intercept provides an estimate of  $\log(\beta)$ . The correlation coefficient  $r$  reflects the quality of the linear fit.
> >
> > > Does each item in the heatmap in Figure 6 represent the same number of demonstrations? The relevant context was difficult to locate in the paper.
> >
> > Each item in the heatmap does not represent the same number of demonstrations. For each unique object in each environment, we collect 120 demonstrations. If there are 4 environments and we collect 2 objects in each, we have  4 * 2 * 120 = 960  demonstrations. If there are 16 environments and we collect 4 objects in each, we have  16 * 4 * 120 = 7,680  demonstrations. Although the number of demonstrations for each item differs, the performance is comparable because, as we have verified in our paper, the number of demonstrations is not the bottleneck limiting performance. We believe that collecting 120 demonstrations for each object is sufficient to saturate the performance growth of almost all policies in terms of the number of demonstrations.
> >
> > > Including an in-domain evaluation would help readers better understand the performance gap between zero-shot evaluation and in-domain data.
> >
> > Thank you for this excellent suggestion. We have added an in-domain evaluation experiment. For the Pour Water task, we collected 200 demonstrations for each of the four environment-object pairs originally used for testing and trained four in-domain policies. We then compared these in-domain policies with the original out-of-domain policy. The results are shown in the table below. It can be seen that by training on a large number of environments and objects, the performance of the out-of-domain policy is already very close to that of the in-domain policy.
> >
> > |              | **In-Domain Policy** | **Out-of-Domain Policy** |
> > | ------------ | -------------------- | ------------------------ |
> > | Score        | 0.953  ±  0.032       | 0.922  ±  0.058           |
> > | Success Rate | 95.0  ±  8.7 %        | 87.5  ±  13%              |
> >
> > > Supplementary suggestion: It would be interesting to explore if the scaling law applies to environments/objects generated by the generative model [1]. Investigating this could reveal a potential gap between synthesized and real-world data and help identify the factors influencing it.
> >
> > Thank you for this fascinating suggestion. Using generative models to obtain a large amount of synthesized data is indeed a simple and efficient way to scale robotic data. We look forward to exploring this avenue in future work to identify potential gaps between synthesized and real-world data and to understand the influencing factors.

---

> ### Comment · Reviewer_facp · 2024-11-25
> **Concerns are well addressed**
>
> Thank you to the authors for the thorough responses. My main concerns, particularly about data consistency, have been effectively addressed. I have updated my score to "accept."

---

### Official Review · Reviewer_GM4F · 2024-10-30

**Soundness:** 3
**Presentation:** 4
**Contribution:** 3
**Rating:** 8
**Confidence:** 3

**Summary:**

The paper investigates the correlation between the size of the dataset and the task performance in behavioral cloning with Diffusion Policy. The authors mainly investigate two manipulation tasks and vary the object to be manipulated and the environment. A key finding of the paper is a power scaling law between the number of training objects / environments and the task performance. Furthermore, the authors highlight that collecting more than 50 demonstrations per environment-object combination does not further increase the performance.

**Strengths:**

The paper is well written and easy to follow. The experiments are clear and the results are presented well. Insights into which data is most useful and how many demonstrations are required to solve a task are certainly very helpful. I particularly like that all experiments are executed on a real robot arm in reasonably cluttered environments.

**Weaknesses:**

The analysis focuses mainly on two relatively simple tasks (pouring water and rearranging a computer mouse). While the paper would benefit from investigating more tasks and also including more challenging tasks, I am aware that it is quite an effort to set up additional experiments with real robots and collect large training sets.

The objects considered in the tasks are always objects of the same category (e.g., water bottles) and, thus, very similar. I believe that this is the reason why increasing the number of objects per environment in Figure 6 does not improve the task performance much. Essentially, the objects are always quite similar, while the environments differ quite significantly, and therefore, it is more important to collect data from more environments rather than with different objects in the same environment.

**Questions:**

1. In Figure 6, sometimes increasing the number of objects per environment decreases the task performance quite considerably (e.g., right plot M=4). This seems quite counterintuitive. What could be the reason for this?

2. What is exactly is the correlation coefficient r (e.g., line 380)?

3. There are no standard deviations in the plots. Consider adding the standard deviations to make it easier to assess the empirical significance of the results.

4. Caption 7: "In the setting where we collect the maximum number of demonstrations, we examine whether the policy's performance follows a power-law relationship with the total number of demonstrations." I believe the first "demonstrations" should be something like "environment-object combinations".

5. Line 47/48: "[W]e aim to investigate the following fundamental question: Can appropriate data scaling produce robot policies capable of operating on nearly any object, in any environment". In my opinion the statement that the paper investigates this fundamental problem is a bit too strong since the policies learn to manipulate only objects of the same category (e.g., water bottles) in environments with similar characteristics (tabletop environments), so these policies are quite far from handling any object in any environment.

---

> ### Author Response · Authors · 2024-11-17
>
> We thank Reviewer GM4F for their insightful and positive review.
>
> > The analysis focuses mainly on two relatively simple tasks (pouring water and rearranging a computer mouse). While the paper would benefit from investigating more tasks and also including more challenging tasks, I am aware that it is quite an effort to set up additional experiments with real robots and collect large training sets.
>
> Thank you for bringing that to our attention. We acknowledge that the tasks investigated in the paper are limited in both number and difficulty. This is mainly due to our use of UMI as the data collection device. The current version of UMI has certain constraints, as discussed in our paper, which prevent us from collecting data on tasks involving occlusions in the camera’s view (e.g., opening doors or drawers) or those requiring very precise actions (e.g., inserting objects). We hope to address these limitations in the future by using a more advanced data collection device that will allow us to validate our data scaling laws on a larger and more complex set of tasks.
>
> > The objects considered in the tasks are always objects of the same category (e.g., water bottles) and, thus, very similar. I believe that this is the reason why increasing the number of objects per environment in Figure 6 does not improve the task performance much. Essentially, the objects are always quite similar, while the environments differ quite significantly, and therefore, it is more important to collect data from more environments rather than with different objects in the same environment.
>
> This is an interesting perspective, and we believe it is quite reasonable. Because objects of the same category are relatively similar in shape, they are easier to generalize over, which is why increasing the number of objects per environment does not significantly improve task performance. We look forward to exploring more challenging generalizations in the future, such as generalizing to novel (but related) object categories.
>
> > In Figure 6, sometimes increasing the number of objects per environment decreases the task performance quite considerably (e.g., right plot M=4). This seems quite counterintuitive. What could be the reason for this?
>
> There could be two reasons for this: (1) Training Randomness: It’s possible that using a different random seed could lead to improved performance. (2) Evaluation Noise and Scoring Bias: The evaluators may have scored this policy lower. We believe that, theoretically, the performance did not decrease, but due to various random factors—especially since we did not perform multiple seed repetitions (complete real-world evaluations are very complex and time-consuming, so we only did one seed)—some scores may have noise. We recommend focusing on the overall trend shown in the figure: as the number of environments increases (e.g., to 16), the performance gap between collecting multiple objects per environment and just a single object becomes negligible.
>
> > What is exactly is the correlation coefficient r (e.g., line 380)?
>
> If two variables $Y$ and  $X$  satisfy the relation  $Y = \beta \cdot X^\alpha$ , they exhibit a power-law relationship. Applying a logarithmic transformation to both $Y$  and $X$  reveals a linear relationship: $\log(Y) = \alpha \log(X) + \log(\beta)$ . Therefore, to determine whether the two variables  $Y$  and $X$  satisfy a power-law relationship, we can transform the original data by taking the logarithm and fit a simple **linear regression** model to the transformed data. We use the correlation coefficient $r$ to assess the quality of the linear fit; if the absolute value of $r$ is close to 1, it indicates that the log-log plot of $Y$  and $X$ fits well with a straight line, suggesting a power-law relationship.
>
> > There are no standard deviations in the plots. Consider adding the standard deviations to make it easier to assess the empirical significance of the results.
>
> Thank you for this valuable suggestion. In Table 1 of the original paper, we reported the standard deviation across 8 unseen environments. We have now uploaded a revised version of the paper, where all the curve plots in the main text include shaded regions representing the 95% confidence intervals to make it easier to assess the empirical significance of the results.
>
> > Caption 7: "In the setting where we collect the maximum number of demonstrations, we examine whether the policy's performance follows a power-law relationship with the total number of demonstrations." I believe the first "demonstrations" should be something like "environment-object combinations".
>
> Yes, your understanding is correct. This setting indeed has the maximum number of “environment-object combinations” (64 combinations) as well as the maximum number of demonstrations, totaling over 6,400. Therefore, we chose this setting to study how changes in the number of demonstrations affect performance.

---

> > ### Author Response · Authors · 2024-11-17
> >
> > > Line 47/48: "[W]e aim to investigate the following fundamental question: Can appropriate data scaling produce robot policies capable of operating on nearly any object, in any environment". In my opinion the statement that the paper investigates this fundamental problem is a bit too strong since the policies learn to manipulate only objects of the same category (e.g., water bottles) in environments with similar characteristics (tabletop environments), so these policies are quite far from handling any object in any environment.
> >
> > Thank you for pointing that out. We indeed did not express this clearly. Your understanding is correct. The core question we aim to investigate is: Can appropriate data scaling produce robot policies capable of operating on nearly any object within the same category, in any environment? We have updated this in the revised version of the paper.

---

> > ### Comment · Reviewer_GM4F · 2024-11-18
> >
> > Thank you for the answers and clarifications.
> >
> > > This is an interesting perspective, and we believe it is quite reasonable. Because objects of the same category are relatively similar in shape, they are easier to generalize over, which is why increasing the number of objects per environment does not significantly improve task performance.
> >
> > I believe the paper would benefit from discussing this briefly.
> >
> > >  There could be two reasons for this: (1) Training Randomness: It’s possible that using a different random seed could lead to improved performance. (2) Evaluation Noise and Scoring Bias: The evaluators may have scored this policy lower. We believe that, theoretically, the performance did not decrease, but due to various random factors—especially since we did not perform multiple seed repetitions (complete real-world evaluations are very complex and time-consuming, so we only did one seed)—some scores may have noise.
> >
> > I see. The stochasticity in the training and evaluation process could explain this phenomenon, and even with these outliers, the trend described in the paper is clearly visible. Nevertheless, I believe that the paper would benefit from an explicit evaluation of how noisy the results are. I am aware that it is not possible to run all these experiments for multiple seeds. Would it be possible to re-run a small subset of the experiments (training and evaluation) multiple times to get a rough estimate of how reliable the scores are? Such an experiment would be very helpful for assessing whether the outliers in Figure 6 are indeed caused by noise.
> >
> > > If two variables $X$ and $Y$ satisfy the relation $Y=\beta \cdot X^\alpha$, they exhibit a power-law relationship. Applying a logarithmic transformation to both $Y$ and $X$ reveals a linear relationship: $\log(Y) = \alpha \log(X) + \log(\beta)$. Therefore, to determine whether the two variables $Y$ and $X$  satisfy a power-law relationship, we can transform the original data by taking the logarithm and fit a simple linear regression model to the transformed data. We use the correlation coefficient to assess the quality of the linear fit; if the absolute value of is close to 1, it indicates that the log-log plot of $Y$ and $X$ fits well with a straight line, suggesting a power-law relationship.
> >
> > So $r$ is the [Pearson correlation coefficient](https://en.wikipedia.org/wiki/Pearson_correlation_coefficient) $\rho$ on the log-transformed data? I believe it would be good to clarify this in the paper.
> >
> > > Thank you for pointing that out. We indeed did not express this clearly. Your understanding is correct. The core question we aim to investigate is: Can appropriate data scaling produce robot policies capable of operating on nearly any object within the same category, in any environment? We have updated this in the revised version of the paper.
> >
> > Thank you for this revision. The revised version more accurately reflects the experiments in the paper.

---

> > > ### Author Response · Authors · 2024-11-21
> > >
> > > > Would it be possible to re-run a small subset of the experiments (training and evaluation) multiple times to get a rough estimate of how reliable the scores are? Such an experiment would be very helpful for assessing whether the outliers in Figure 6 are indeed caused by noise.
> > >
> > > Thank you for this insightful suggestion. We conducted an additional experiment to assess the reliability of our scores. Specifically, for the Mouse Arrangement Task, we retrained the original 4 policies (with  N/M = 4, 3, 2, 1 and  M = 4 ) using two new random seeds, resulting in a total of 12 policies. These policies were evaluated in 4 unseen environments, each containing one novel object, with 5 trials per environment.
> > >
> > > The results are presented in the table below:
> > > | Number of Objects Per Environment (N/M) | Seed 1 | Seed 2 | Seed 3 | Mean ± STD    |
> > > | --------------------------------------- | ------ | ------ | ------ | ------------- |
> > > | 4                                       | 0.65   | 0.592  | 0.475  | 0.572 ± 0.073 |
> > > | 3                                       | 0.608  | 0.517  | 0.608  | 0.578 ± 0.043 |
> > > | 2                                       | 0.575  | 0.6    | 0.583  | 0.586 ± 0.01  |
> > > | 1                                       | 0.583  | 0.592  | 0.625  | 0.6 ± 0.018   |
> > >
> > > As the table illustrates, there is some variance between different training seeds. However, the overall means show that the performance gap between collecting different numbers of objects is minimal. This finding supports our assertion that the outliers observed in the original Figure 6 were indeed caused by noise.
> > >
> > > > So $r$ is the Pearson correlation coefficient $\rho$ on the log-transformed data?
> > >
> > > Yes,  $r$  is the Pearson correlation coefficient  $\rho$  calculated on the log-transformed data.

---

> > > > ### Comment · Reviewer_GM4F · 2024-11-22
> > > >
> > > > > Thank you for this insightful suggestion. We conducted an additional experiment to assess the reliability of our scores. Specifically, for the Mouse Arrangement Task, we retrained the original 4 policies (with N/M = 4, 3, 2, 1 and M = 4 ) using two new random seeds, resulting in a total of 12 policies. These policies were evaluated in 4 unseen environments, each containing one novel object, with 5 trials per environment.
> > > >
> > > > Thank you for this additional experiment. However, the results are not entirely clear to me. You said that you ran the experiment with two new seeds. So one of the three seeds that you show is the one that you already used in the paper, right? Then I would expect that one of the columns shows the same results as Figure 6, right, middle column. Yet, all columns show values different from those in the paper. Did you change anything else in the evaluation?
> > > >
> > > > > Yes, $r$ is the Pearson correlation coefficient calculated on the log-transformed data.
> > > >
> > > > Thanks for the clarification. You could mention that explicitly in the paper to make this clear.

---

> > > > > ### Author Response · Authors · 2024-11-22
> > > > >
> > > > > > Thank you for this additional experiment. However, the results are not entirely clear to me. You said that you ran the experiment with two new seeds. So one of the three seeds that you show is the one that you already used in the paper, right? Then I would expect that one of the columns shows the same results as Figure 6, right, middle column. Yet, all columns show values different from those in the paper. Did you change anything else in the evaluation?
> > > > >
> > > > > Yes, your understanding is correct. The policy corresponding to Seed 1 is the same one used in the paper. The policies corresponding to Seed 2 and Seed 3 were newly trained. The discrepancy in values arises because we slightly simplified the evaluation setting for this experiment. In the original paper, each policy was evaluated on 8 unseen environments. Here, however, each policy was evaluated on only 4 unseen environments.
> > > > >
> > > > > It is also worth noting that even if we were to evaluate on 8 unseen environments now, it is unlikely that we would obtain the exact same scores as reported in the original paper. This is due to two factors: (1) the environments have undergone minor changes since the earlier experiments (but when comparing different policies from the same batch for the paper, we ensured the environments remained consistent); and (2) the subjective scoring by evaluators introduces some degree of variance.

---

> > > > > > ### Comment · Reviewer_GM4F · 2024-11-22
> > > > > >
> > > > > > Alright, thanks for the clarification. I was not aware that you repeated the evaluation also for the first seed. In that case, it makes sense that the scores changed.

---

### Official Review · Reviewer_xsiU · 2024-10-31

**Soundness:** 3
**Presentation:** 4
**Contribution:** 3
**Rating:** 8
**Confidence:** 5

**Summary:**

The authors in this work construct zero-shot generalizable policies with imitation learning and propose some scaling rules that determines the performance of such policies with the amount of data that is used to train such policies. Using such predicted scaling laws, the authors then collect training data for two new tasks, and show that the laws hold for such tasks as well.

**Strengths:**

The authors present a compelling paper with a hypothesis that until very recently were not widely known in the field of robot learning. To summarize the strengths of the paper:

1. The focus of this paper, behavior cloning models that generalize to new objects and environments zero shot, has only very recently been shown to work, so analyzing the underlying principles that determine the success of this process is very important.
2. The authors approach this problem in a principled way, collecting a lot of data, training a number of models, and evaluating on those model in the real world over different environments is a hard but important problem.
3. The scaling law proposed by the authors seem to hold up in further examinations with two different tasks trained post-facto.
4. The visual presentation of the objects and scenes where data is collected is done wonderfully.
5. The extra tips in the appendix for practitioners for training more general policies seem to be also very helpful for further research in this direction.

**Weaknesses:**

While the paper is advancing robot learning in a positive directions, there are possible improvements that can be made. For example:

1. The primary issue with the paper is that it does not mention the initial conditions for the robot and the environment – and how they were varied. For example for the pouring task, it is unclear whether the cup and the red dot is located at the same relative position to the bottle, and if so, if it's sufficient for the robot to open-loop follow a training trajectory afterwards to complete the task. To understand what the authors mean by "90% success", we must have a good sense of what the task and environment variations look like.

2. Similarly, the authors don't create much of a variation in the task difficulty – as a result it is hard to tell if the scaling laws derived in the paper scales in a similar way with harder or easier tasks.

3. The "new tasks" that the authors collect data for and evaluate in (towel folding, unplug charger) does not seem to be of a similar level of difficulty as the primary tasks talked about in the paper.

**Questions:**

1. Why do we see the normalized score drop at the very end as the number of demonstrations are scaled?
2. What level of variation is done in the evaluation environment setup over multiple runs in the same environment on the same object?

---

> ### Author Response · Authors · 2024-11-17
>
> We thank Reviewer xsiU for their insightful and positive feedback.
> > The primary issue with the paper is that it does not mention the initial conditions for the robot and the environment. To understand what the authors mean by "90% success", we must have a good sense of what the task and environment variations look like.
>
> Thank you for bringing this up. In fact, in Appendix D: Task Details, we provide a detailed description of the initial conditions for the environment and objects for each task. Generally speaking, during every evaluation, we command the robot to move to the same initial joint position (primarily for ease of testing), but there are significant variations in the environment and objects. Firstly, to test generalization, our environments or objects are ones that the policy has never seen during training. Secondly, the initial positions of the objects vary greatly, covering almost the entire workspace within the robot's kinematic reach. Below, we provide descriptions of the initial conditions for each task:
>
> 1. Pour Water: The bottle is randomly placed on the table, provided it is within the robot’s kinematic reach.  The relative initial position of the bottle and mug is also randomized, ensuring they are spaced variably while keeping the mug visible to the camera after the bottle is grasped. Only the red coaster has limited initial conditions; it is a 9 cm diameter circle consistently placed approximately 10 cm to the right of the mug, used across all environments.
> 2. Mouse Arrangement: The mouse can be positioned anywhere on the table, as long as it remains within the robot’s kinematic reach. The mouse may be oriented straight ahead, in which case the robot needs to grasp it directly from behind. Alternatively, it might be slightly tilted to the left or right, necessitating the robot to employ non-prehensile actions, such as pushing the mouse into the correct orientation before closing the gripper for picking it up. In each test, the relative position between the mouse pad and the mouse is also varied, with the mouse pad randomly placed around the mouse.
> 3. Fold Towels: The initial position of the towel may vary on the table, provided it remains within the robot’s kinematic reach and its tilt angle relative to the table’s edge does not exceed 15 degrees. We assume the towel has already been folded several times.
> 4. Unplug Charger: The charger and power strip can be placed anywhere on the table, provided they remain within the robot's kinematic reach.
>
> > The authors don't create much of a variation in the task difficulty – as a result it is hard to tell if the scaling laws derived in the paper scales in a similar way with harder or easier tasks.
>
> Thank you for pointing this out. There are indeed some differences in difficulty among the tasks we have selected. For example, Pour Water is a relatively long-horizon task and is more challenging than Mouse Arrangement. However, we did not use even more difficult tasks mainly because the current version of UMI has certain constraints, as discussed in our paper, which prevent us from collecting data on tasks involving occlusions in the camera’s view (e.g., opening doors or drawers) or those requiring very precise actions (e.g., inserting objects). We hope to address these limitations in the future by using a more advanced data collection device that will allow us to validate our data scaling laws on a larger and more complex set of tasks. We believe that the trends regarding data scaling laws also hold true for extremely hard tasks.
>
> > The "new tasks" that the authors collect data for and evaluate in (towel folding, unplug charger) does not seem to be of a similar level of difficulty as the primary tasks talked about in the paper.
>
> The reason for not selecting more difficult "new tasks" is the same as mentioned above. We look forward to utilizing better data collection device in the future to validate our data scaling laws on a larger and more diverse set of tasks.
>
> > Why do we see the normalized score drop at the very end as the number of demonstrations are scaled?
>
> Thank you for pointing this out. We believe there are several possible reasons: (1) The model size might not be sufficiently large, lacking enough capacity to fit the increased data volume. (2) The training time might be insufficient, meaning the model has not fully converged. (3) Training randomness—it's possible that a different random seed could lead to improved performance. (4) Evaluation noise and scoring bias are unavoidable. Given that a full evaluation is time-consuming (see our next response), we have yet to determine the precise cause. We lean towards believing that the theoretical performance of the model remains unchanged, and that the slight drop in score may be attributed to noise during training and testing.

---

> > ### Author Response · Authors · 2024-11-17
> >
> > > What level of variation is done in the evaluation environment setup over multiple runs in the same environment on the same object?
> >
> > In our first response, we described the variations in object positions during evaluation. Here, we provide an example of our entire evaluation workflow using the environment generalization experiment. Recall that in this experiment, we collect data with the same object across 32 environments, training a total of 21 policies. Each policy is evaluated in 8 unseen environments using the same object, with 5 trials per environment. The average normalized score from these 40 trials is reported for each policy.
> > Operationally, we complete the training of all 21 policies before deploying the entire robotic system into a new environment for evaluation. To ensure unbiased results, we conduct blind tests: initially, we set an initial position for the object, as described in our first response, then randomly shuffle the order of the 21 policies. Testing then proceeds at this initial position, with the 21 shuffled policies scored according to the criteria outlined in Appendix D. Subsequently, we select a new initial position for the object and repeat the scoring process for the 21 shuffled policies. This procedure is replicated five times to conclude the testing in one environment. The robot system is then transitioned to another new environment, and the entire process is repeated, completing eight cycles in total for all tests.

---

### Meta-Review · Area_Chair_JGue · 2024-12-19

**Metareview:**

The paper investigated data scaling laws in the case of behavior cloning in robot manipulation. They specifically try diffusion policy as the algorithm and UMI as the data collection device. During this work, many demos were collected in real (40K), and many real robot roll-outs were tested (15K), based on which power-law relationships were established between the data and model performance.

The reviewers agree that the paper is well-written, the experiments are well-executed, and the work can inform future research. The AC shares this view and would like to recommend the paper be accepted. The AC would suggest that the paper incorporates the suggestions made by the reviewers in clarifying various parts for the final version.

**Additional Comments On Reviewer Discussion:**

The paper was well-received even before the rebuttal. Reviewers asked many clarificatory questions about the details of the work, and the authors responded well to those. Appropriately, some reviewers further increased their scores.

---

### Decision · Program_Chairs · 2025-01-22

Accept (Oral)